# tgCRISPRi: efficient gene knock-down using truncated gRNAs and catalytically active Cas9

Ankush Auradkar [1], Annabel Guichard[1], Saluja Kaduwal[1], Marketta Sneider[1] & Ethan Bier [1,2] ✉

CRISPR-interference (CRISPRi), a highly effective method for silencing genes in mammalian cells, employs an enzymatically dead form of Cas9 (dCas9) complexed with one or more guide RNAs (gRNAs) with 20 nucleotides (nt) of complementarity to transcription initiation sites of target genes. Such gRNA/dCas9 complexes bind to DNA, impeding transcription of the targeted locus. Here, we present an alternative gene-suppression strategy using active Cas9 complexed with truncated gRNAs (tgRNAs). Cas9/tgRNA complexes bind to specific target sites without triggering DNA cleavage. When targeted near transcriptional start sites, these short 14–15 nts tgRNAs efficiently repress expression of several target genes throughout somatic tissues in *Drosophila melanogaster* without generating any detectable target site mutations. tgRNAs also can activate target gene expression when complexed with a Cas9-VPR fusion protein or modulate enhancer activity, and can be incorporated into a gene-drive, wherein a traditional gRNA sustains drive while a tgRNA inhibits target gene expression.

Adaptation of CRISPR-Cas9 defense systems in bacteria for gene editing purposes in diverse prokaryotic and eukaryotic cells has revolutionized nearly every field of biology in the past few years. The most broadly used synthetic dual-component system consists of Cas9 (CRISPR-associated protein) endonuclease and programmable guide RNAs (gRNAs) that direct Cas9 to cleave specific DNA target sites[1–3]. In its native form, catalytically active Cas9 complexes with a gRNA to bind a specific 20-long complementary DNA target sequence and induce a double-strand break (DSB)[1].

Modified versions of the binary Cas9/gRNA CRISPR system can also be employed to regulate expression of specific genes enabling strategies to control both DNA information content and RNA expression[4]. Mutating residues critical for the activity of both the RuvC and HNH nuclease domains of Cas9 renders it catalytically inactive (dead Cas9 or dCas9), permitting its gRNA-dependent sequence-specific DNA binding activity to be repurposed for genetic perturbation beyond DNA cleavage[5–15]. Recent studies have shown that dCas9 can be

fused to different protein domains that suppress or activate gene expression or that change the epigenetic state of the targeted locus[5–9]. In one such system, known as CRISPR interference (CRISPRi), dCas9 is combined with a transcriptional repression domain such as KRAB. dCas9-KRAB/gRNA complexes are recruited to transcriptional start sites (TSS) of endogenous genes, as specified by one or more gRNAs, to repress transcription of a targeted gene[8]. In the CRISPR activation (CRISPRa) system, one or more transcriptional activators are recruited to bind sequences upstream of a TSS using a single targeting gRNA to over- or mis-express the locus. Various transcriptional activator domains are either directly fused to dCas9 (e.g., VP64, p65, and RTA in the VPR system) or recruited to an engineered gRNA system through a minimal aptamer appended to gRNA that can bind to the dimerized MS2 bacteriophage coat protein fused to various transcriptional activators[11–15].

Other studies have shown that if the gRNA is truncated (tgRNA), targeting a ≤16-nt-long sequence, it retains its ability to direct Cas9

[1]Department of Cell and Developmental Biology, University of California, San Diego, 9500 Gilman Drive, La Jolla, CA 92093-0335, USA. [2]Tata Institute for Genetics and Society - UCSD, La Jolla, USA. ✉e-mail: ebier@ucsd.edu

binding to specific genomic DNA targets, but does not result in DNA cleavage[4,16–18]. Further modified Cas9 forms fused to activation or repression domains and engineered tgRNAs have been employed to achieve transcriptional activation of endogenous genes or repression in mammalian cells[16,19,20]. Truncated gRNAs have also been implicated in regulating gene expression in bacteria[18,21–23]. For example, a recent study on the *Streptococcus pyogenes* CRISPR-Cas9 system revealed that a natural, single 11-nt-long gRNA could direct Cas9 to repress transcription from its own promoter[23].

Given its inherent simplicity and versatility, we felt that the relatively under-exploited alternative strategy of employing fully active Cas9 to sustain both gene editing (with regular gRNAs) and gene expression control (with tgRNAs) deserved further assessment. In particular, the use of tgRNAs has not been examined in insects where it could offer great potential as a scarless strategy for modulating gene activity in the context of improving the performance or expanding the functionality of gene-drive systems.

In this study, we designed a CRISPRi system based on tgRNAs of 14–16-nt-long complementarity and active Cas9. We tested tgRNAs targeting several *Drosophila* loci and found that they can efficiently silence gene expression. tgRNAs targeting sequences at the TATA box just upstream of a TSS or the TSS itself produces loss-of-function (LOF) phenotypes similar to those achieved by active Cas9 cleaving and mutating target genes using full-length gRNAs, and can reduce target gene expression nearly as effectively as dCas9 bound to full-length gRNAs. We also show that tgRNAs can be used in an activating mode (i.e., CRISPRa) and that tgRNAs targeting cis-regulatory modules (or enhancers) similarly can be employed either to reduce or to activate gene expression modulating tissue-specific expression of essential genes. This study thus validates the effectiveness of the truncated gRNA CRISPR interference system (tgCRISPRi) in *Drosophila*. Such dual action systems are of particular advantage when one is constrained to use an active form of Cas9 at the same time, as exemplified by CRISPR-based gene-drive systems. With this important application in mind, we provide proof-of-principle for incorporating tgCRISPR technology into a gene-drive, wherein a full-length gRNA promotes copying of a gene-drive element while a tgRNA simultaneously modulates specific gene expression. We discuss how tgRNAs can expand the range of CRISPR applications by enabling multiple parallel gene-editing and gene-modulatory functionalities across diverse gene editing applications.

## Results

### tgCRISPRi achieves efficient transcriptional repression of endogenous genes in *Drosophila*

Previous studies have demonstrated that Cas9-VPR/tgRNA complexes can sustain sequence-specific binding and transcriptional activation of target genes[16,17,19]. Based on this observation, we hypothesized that tgRNAs bound to fully active native Cas9 (tgCRISPRi) might similarly repress the expression of endogenous genes in insects such as *Drosophila* and could be used in combination with full-length gRNAs to perform genome editing or augment the functionality of gene-drive systems used to combat vector borne diseases or crop pests. As an initial test of this hypothesis, we developed a tgCRISPRi system to suppress transcription of two classic pigmentation loci, *yellow* (*y*) and *ebony* (*e*), as revealed by visible body color phenotypes. The *y* locus, located at the tip of the X chromosome, is a commonly employed site for genome engineering studies[24–26]. *y* is expressed in localized patterns, including developing epidermis, wings, bristles, larval mouthparts, and denticle belts, where it is required to produce the dark body pigment melanin. Disruption of *y* gene function leads to complete elimination of black pigment, resulting in a yellow cuticle color[27–30]. Conversely, the *e* gene, which is not spatially regulated, acts to suppress melanin formation. *e* mutants exhibit the opposite phenotype of overall dark pigmentation[31].

We designed pairs of tgRNAs (14–15-nt long) and canonical full-length gRNAs (20-nt long) targeting the same sites (Fig. 1a), located in a window between −120 bp and +20 bp of the respective TSSs of both the *y* and *e* genes (Fig. 1b), and generated transgenic flies expressing pairs of *tgRNAs* or corresponding full-length *gRNAs* driven by the *U6:3* and *U6:1* promoters[32]. We tested these *tgRNA* and full-length *gRNA* lines with a *vasa-Cas9* transgene as a source of Cas9 since it is expressed ubiquitously at low levels in somatic cells as well as at higher levels in the germline[33]. Adult flies expressing Cas9 under the *vasa* promoter and full-length 20 nt *gRNAs* produced expected loss-of-function (LOF) pigmentation phenotypes for both *y* and *e*[32]. It is noteworthy that transgenic strains (*vasa-Cas9*, *gRNAs*, and *tgRNAs*) carry duplicate sets of the *y*+ target gene, since the attP docking site for the Cas9-bearing cassette is marked with a *y*+ transgene (Fig. 1c). Thus, production of the yellow phenotype requires four copies of the *y* gene to be mutated simultaneously. In the case of *e*, the full-length *gRNAs-e(1 + 2)*, when co-expressed with *vasa-Cas9* produced an increase in black pigmentation, which is typical of the *ebony* LOF phenotype (Fig. 1c). Although *vasa-Cas9* is expressed ubiquitously, we detected mosaic pigmentation phenotypes with both *gRNAs-y(1 + 2)* and *gRNAs-e(1 + 2)* producing patches of *y*+ and *e*+ tissue, respectively. These wild-type clones presumably derived from non-homologous end joining (NHEJ) mutational events resulting in functional alleles resistant to further cleavage or to rare intact *y*+ and *e*+ alleles.

Mirroring the results with mutagenic full-length gRNAs, all adult flies co-expressing pairs of *tgRNAs* targeting the *y* and *e* loci with *vasa-Cas9* displayed strong and fully penetrant LOF pigmentation phenotypes covering the entire body. Moreover, flies expressing *vasa-cas9 and tgRNAs-e(1 + 2)* exhibited considerably darker coloration of the cuticle throughout the body than observed with the full-length gRNA pair. This strong uniform phenotype suggests that *tgRNAs-e(1 + 2)* resulted in highly efficient biallelic loss of *e* function in somatic cells (Fig. 1c). Similarly, in the case of *tgRNAs-y(1 + 2)*, for which 4 copies of the *y* gene must be silenced, all flies displayed full body yellow cuticles with no signs of mosaicism (Fig. 1c). As expected, we observed similar strong LOF phenotypes when using an enzymatically dead Cas9 (dCas9) with either pair of full-length gRNAs or tgRNAs targeting the *y* and *e* loci (Fig. 1c). These observations indicate that paired tgRNAs of 14–15-nt length targeting a window between −120 and +20 bp of the TSS can lead to strong penetrant LOF phenotypes when combined with fully active Cas9, comparable to those achieved with full-length gRNAs either by silencing using dCas9 or by mutating the target loci with active Cas9.

### CRISPR-mediated gene repression occurs via distinct mutagenic (gRNA) versus non-mutagenic (tgRNA) mechanisms

Having produced efficient and penetrant of LOF phenotypes with both full-length gRNAs and shortened tgRNAs at the *y* and *e* loci, we next examined whether these mutant phenotypes were associated with induction of target site mutations. Typically, Cas9/gRNA complexes create DSBs that eventuate in mutations generated by the error-prone NHEJ repair pathway[34]. We confirmed this prediction by crossing F1 males carrying both *vasa-cas9* and full-length *gRNAs* to reference *y or e* mutants and then scoring F2 offspring for germline transmission of *y or e* mutations. We found that approximately half of the F2 progeny from F1 *gRNAs-e(1 + 2)/Y; vasa-cas9* males showed full-bodied *e* mutant phenotypes (Fig. 2a). Similarly, we crossed *gRNAs-y(1 + 2)/Y; vasa-cas9* F1 males to *y* mutant females. In this case, the result was sterility: females laid only a few eggs which did not hatch. Since LOF *y* mutants are viable and fertile, this male-derived sterility suggests that Cas9/*gRNAs-y(1 + 2)* may have induced multiple chromosome breaks leading to deleterious chromosomal rearrangements. Furthermore, because *Cas9/gRNAs-y(1 + 2)* males have three copies of the *y* gene (the endogenous locus on the chromosome-X, one copy linked to the Cas9

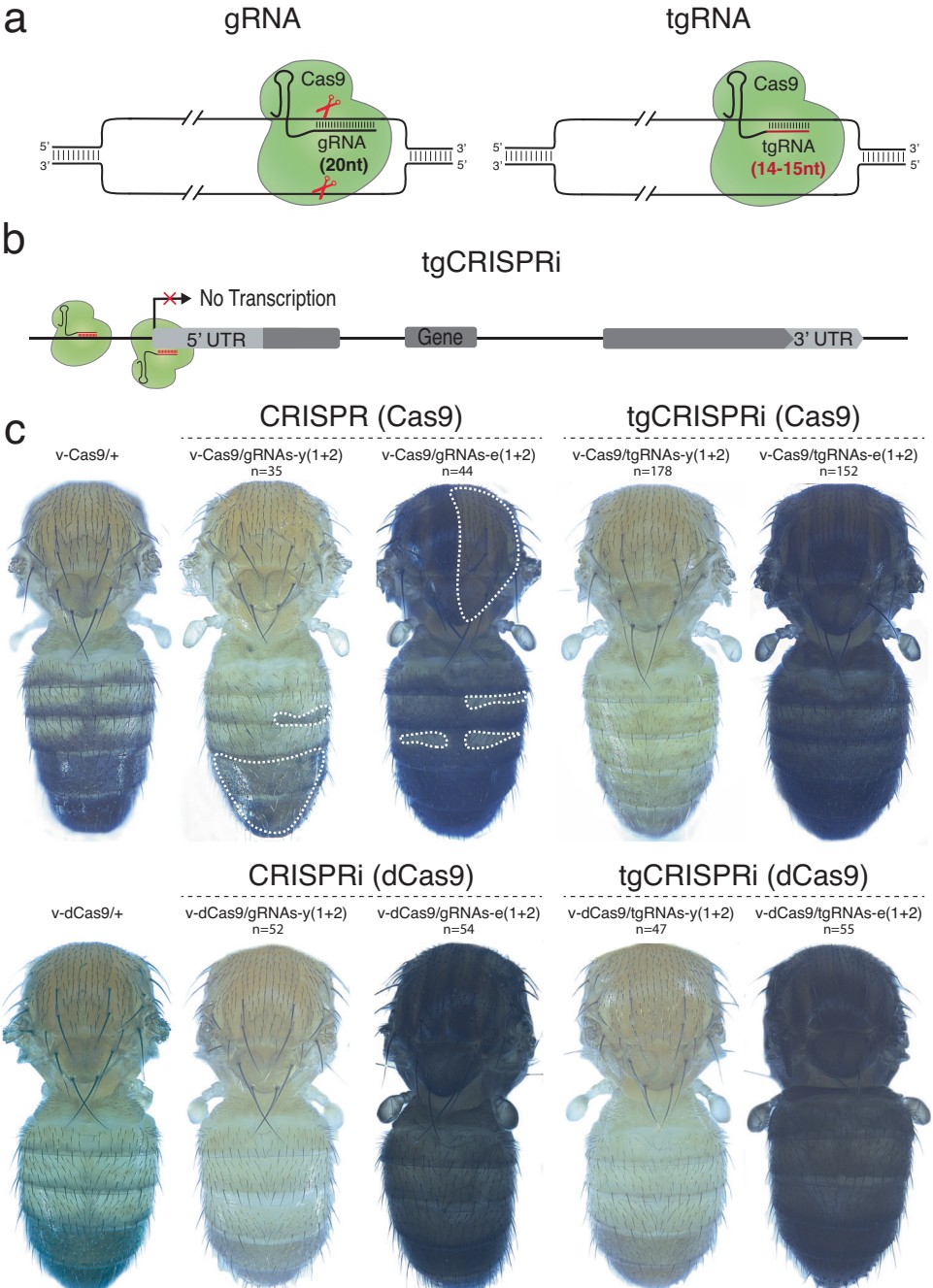

**Fig. 1 | tgRNAs can mediate gene repression using active Cas9 in *Drosophila*.**
**a** Illustration of the gRNA and tgRNA systems. Catalytically active Cas9 protein (Cas9, green) is directed to a particular DNA sequence based on the 20 nt targeting sequence of the gRNA (gray) or 14–15 nt targeting sequence of the tgRNA (red) and an adjacent PAM sequence. Both Cas9:gRNA and Cas9:tgRNA complexes bind to the target sequence; however, only full-length gRNA introduces DSBs and results in genome editing. **b** Schematic representation of the truncated guide CRISPR interference (tgCRISPRi) system. The *Drosophila* gene locus shows relative positions of the targeting tgRNAs. The transcription start site is marked by an arrow while a solid gray box shows the 5′UTR, exon, and 3′UTR of the gene locus. A pair of tgRNAs located upstream and downstream of the transcription start site (TSS) was tested for transcriptional knock-down. Binding of the Cas9:tgRNA complex upstream of the TSS interferes with transcription initiation by preventing RNA polymerase recruitment, while its assembly at a downstream site prevents transcription elongation. **c** Comparison of CRISPR, CRISPRi, and tgCRISPRi system for target editing and repression, respectively, in *Drosophila*. *y* and *e* genes were targeted using *vasa-Cas9* or *vasa-dCas9* and a pair of full-length *gRNAs* or *tgRNAs* produced highly penetrant pigmentation defects in whole flies (wings and legs were removed for observation). The dotted lines represent $y^+$ or $e^+$ patches. *n* = number of samples.

transgenic element on chromosome-3R, and another copy marking the attP target site into which the *gRNAs-y(1 + 2)* transgenic element is inserted on chromosome-3L), a significant proportion of their cells may carry unrepaired or incorrectly repaired DSBs leading to defects in germline development (Fig. 2a). Alternatively, increased DNA repair activity may interfere with processes required for male fertility. Future studies may clarify this question. In contrast to the mutagenic outcomes observed with full-length gRNAs, when *tgRNAs-y/e(1+2)*; *vasa-cas9* F$_1$ males were crossed to *y* or *e* mutant females, 100% of F$_2$ offspring displayed wild-type $y^+$ or $e^+$ phenotypes, respectively, indicating that no detectable LOF germline mutations were produced at either of these loci (Fig. 2a).

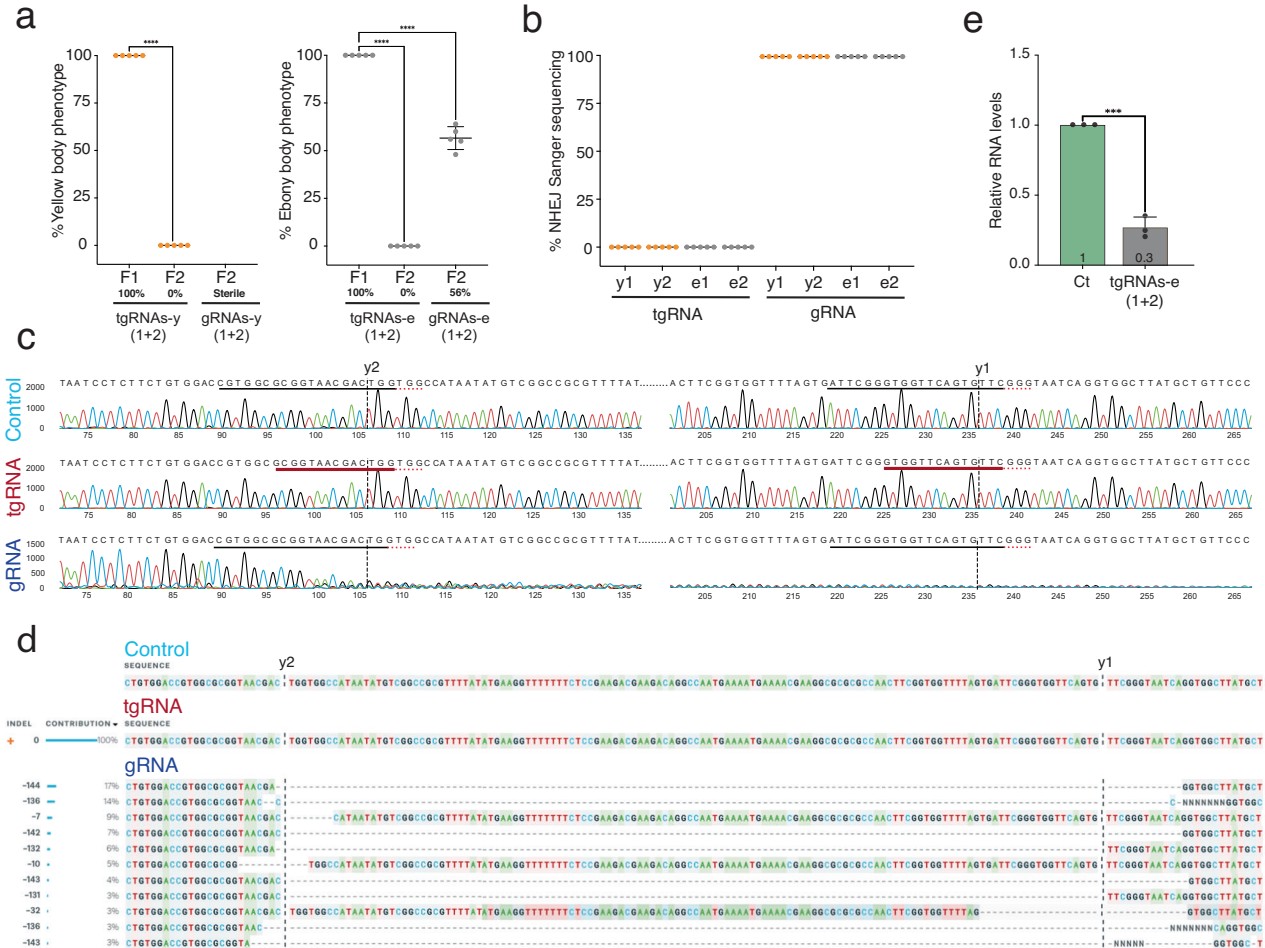

**Fig. 2 | Repression versus cutting of endogenous genes using tgCRISPRi and CRISPR in *Drosophila*. a** Graph shows comparison of body color phenotypes in $F_1$ and $F_2$ progeny generated by crossing $F_1$ males carrying the *vasa-Cas9* source and either tgRNAs or full-length gRNAs to females homozygous for reference *y* or *e* mutations ($n = 5$ individual crosses). Data were analyzed using One-way ANOVA. Data plotted as mean ± standard deviation, ****$p < 0.0001$. **b** Graph shows comparison of NHEJ events generated by either tgRNA or gRNA in $F_1$ progeny. Data were generated using the ICE tool ($n = 5 \times 12$ flies per target gene). **c** Comparison of CRISPR and tgCRISPRi DNA sequence Chromatograms reveals differential generation of NHEJ events at *y* gene target sites. The control *y* + DNA sequence around the tg/gRNA-y2 and tg/gRNA-y1 target sites is shown on top of the 1st and 4th row, respectively. Chromatograms for control (+/vasa-Cas9) animals (1st row) show no NHEJ events revealed as double peaks. Upon introduction of tgRNA-y2 and y1,

peaks appear similar to the control (2nd row). In gRNA-y2 or y1 expressing animals (3rd row), however, multiple peaks appear around the target sites, confirming double-strand cleavage and imprecise DNA repair. **d** Detection of tgCRISPRi and CRISPR-induced mutations for guides targeting the *y* gene. The percentage of indels was quantified using the Synthego ICE online tool. $F_1$ progeny from control, tgRNAs, and gRNAs master flies were sequenced. For controls and tgRNAs, 100% of $F_1$ alleles were wild type. For full-length gRNAs, however, 100% of $F_1$ alleles were NHEJ events. Sequences on the right side show the mutations with the sizes of deletions (−) and insertions (+), with the percentage of indels and KO score for each allele. **e** RT-qPCR of *e* gene from control (Ct = +/*vasa-Cas9*) and tgRNA-e(1 + 2) (*tgRNA/vasa-Cas9*) adult flies. Transcript levels of *rp49* were used as a reference. Error bars represent standard deviation of the mean of three independent experiments and *P* values derived using a two-tailed unpaired T-test, ****$p < 0.0001$.

We complemented our phenotypic assessment of full-gRNA versus tgRNA outcomes by genomic sequencing analysis (using the ICE tool) of regions encompassing their target sites in individual flies to detect Cas9-mediated mutagenic events (Fig. 2b–d). We observed that full-length gRNAs when combined with the *vasa-Cas9* source resulted in the production of indels (insertions and deletions) which often deleted sequences between the two gRNAs. Furthermore, DNA sequencing chromatograms from both *y* and *e* target sites revealed that 100% of target sequences carried mutations at both gRNA target cleavage sites (Fig. 2b–d and Sup. Figs. 1, 2). In contrast, combination of tgRNAs with *vasa-Cas9* did not produce detectable levels of indels at either the *y* or *e* loci (Fig. 2b–d and Sup. Figs. 1, 2). These results are consistent with our hypothesis that the LOF phenotypes observed with full-length gRNAs result from DSBs induced by Cas9, while those induced by tgRNAs arise most likely through non-mutagenic repression of target gene transcription. This latter effect of tgRNAs is

presumably due to tgRNA/Cas9 complexes binding near TSS, precluding access by RNA-polymerase complexes to critical DNA sequences required for initiating transcription (Sup. Fig. 3a, b). We further tested this hypothesis by performing RT–qPCR experiments to quantify the relative transcripts level of the *e* gene in adult flies. We observed that in adult *tgRNAs-e(1 + 2)/Cas9* flies, expression of the *e* gene was significantly reduced relative to control flies (Fig. 2e), indicating that the tgRNA/Cas9 complexes binding DNA near TSS greatly reduced transcription of this target gene. Because full-length gRNAs and tgRNAs targeting *e* produced similar strong LOF pigmentation phenotypes, we quantified the efficiency with which the full-length gRNAs versus tgRNAs reduced transcription of *e* by performing RT-PCR on *tgRNAs-e(1 + 2)/dCas9* and *gRNAs-e(1 + 2)/dCas9* adult flies. We observed that full-length gRNAs-e(1 + 2) reduced transcript level by 90%, while tgRNAs-e(1 + 2) reduced levels by 80% (Sup. Fig. 4). We conclude that tgRNAs combined with Cas9 can efficiently reduce

target gene expression without generating any detectable target site mutations.

## Single tgRNAs can be sufficient for Cas9-dependent repression

Our initial experiments testing the tgCRISPRi strategy employed pairs of tgRNAs to increase the likelihood of success. We further investigated whether both tgRNAs were required for the observed effects or if single tgRNAs alone could generate fully penetrant LOF phenotypes. We therefore generated transgenic fly lines expressing single tgRNAs from the U6:3 promoter[32]. In the case of *y*, *tgRNA-y1* and *tgRNA-y2* targeted positions +1 and −129 relative to the TSS, respectively (Fig. 1c and Fig. 3a). Analysis of the effect of each of these two tgRNAs separately revealed that repression activity of the pair could be attributed entirely to *tgRNA-y1*, which presumably binds and blocks access to initiator sequences of the *y* gene promoter (Fig. 3a). In the case of the *e* gene, *tgRNA-e2* targets a TATA box sequence at position −12, while *tgRNA-e1* targets a site at +21 relative to the TSS (Fig. 3b). We found that repression activity of the pair was provided solely by *tgRNA-e2* (Fig. 3b). RT−qPCR experiments confirmed a significant decrease in *e* transcript levels in flies expressing Cas9 and *tgRNA-e2*, but not with *tgRNA-e1* (Sup. Fig. 5a). Thus, a single tgRNA is sufficient to mediate effective repression when combined with fully active Cas9. In addition, our results indicate that regions between the TATA box and the TSS initiator are good targets for efficient gene repression by a tgRNA.

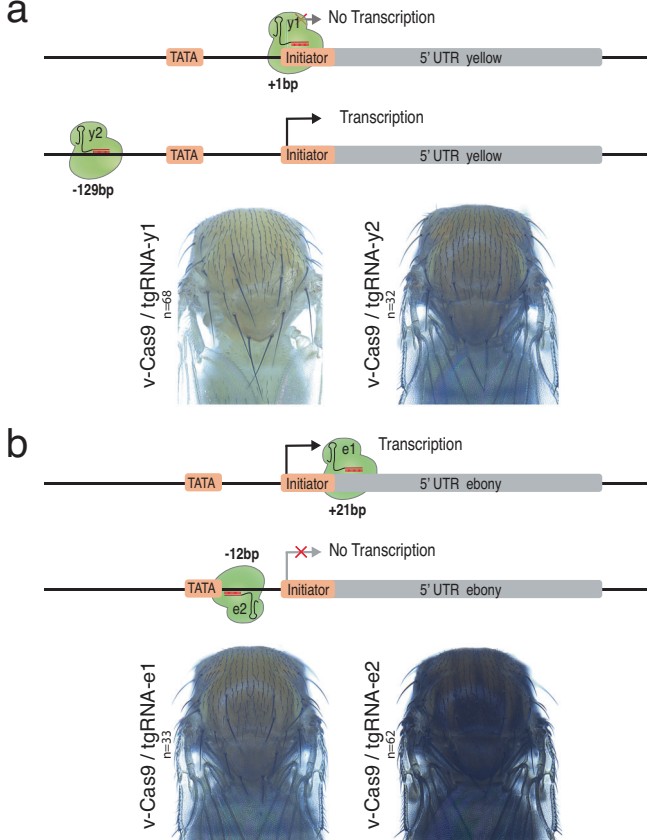

**Fig. 3 | Single tgRNAs can sustain effective and specific tgCRISPRi. a** Genomic representation of the *y* locus showing the position of the tgRNAs designed relative to the TSS. The coordinates of the tgRNA positions are indicated relative to the TSS. Targeting the *y* gene using *vasa-Cas9* and *tgRNA-y1* or *y2* results in pigmentation defects in flies. **b** Genomic representation of the *e* locus showing the position of the tgRNAs relative to the TSS. The coordinates of the tgRNA positions are indicated based on TSS. Targeting the *e* gene using *vasa-Cas9* and *tgRNA-e1* or *e2* results in pigmentation defects in flies. *n* = number of samples.

## tgCRISPRi can also efficiently repress non-TATA box genes

Experiments described above with tgRNAs targeting the *y* and *e* loci demonstrate that tgCRISPRi can efficiently repress expression of TATA box-containing genes. We further investigated whether such tgRNA-mediated repression also could be applied to genes with TATA-less promoters. We generated transgenic strains expressing pairs of tgRNAs targeting a window between −120 and +20 nts of the TSSs of *white* (*w*) and *wingless* (*wg*), two genes lacking a promoter proximal TATA box (Fig. 4a, c). The X-linked *w* gene encodes a subunit of an ATP-binding cassette (ABC) transporter, required for deposition of metabolites into pigment cells of compound eyes, ocelli, Malpighian tubules, and testes[35]. The autosomal *wg* gene encodes a signaling ligand, which, as its name suggests, is required for the formation and patterning of the adult fly wing. In addition, *wg* exerts several other essential functions throughout *Drosophila* development[36,37].

Adult flies expressing *tgRNA-w1-2* and *Act5C-Cas9* displayed LOF phenotypes in which eye coloration is reduced or entirely absent. In the case of *tgRNAs-w(1 + 2)*, we observed substantial and uniform loss of pigmentation across the entire eye in all *Cas9/tgRNAs-w(1 + 2)* flies, indicative of greatly reduced *w* gene expression (Fig. 4b). Next, we analyzed the effect of each of these two tgRNAs separately and found that only tgRNA-w1, which binds to TSS sequences of the *w* gene promoter, displayed strong repression activity (Fig. 4b). In addition, we performed RT−qPCR experiments to quantify relative levels of *w* gene expression in adult flies. Both tgRNA-w1 and tgRNA w2 reduced *w* transcripts, however, the reduction with *tgRNA-w1* was more pronounced, consistent with *tgRNA-w1* exhibiting a stronger LOF eye pigmentation phenotype, comparable to that observed in *tgRNAs-w(1 + 2)* flies (Sup. Fig. 5b).

In the case of the *tgRNAs-wg(1 + 2)* and *Act5C-Cas9* combination, we observed a penetrant lethal phenotype during pupal stages. When combined with the weaker *vasa-Cas9* source, *tgRNAs-wg(1 + 2)* produced viable progeny displaying the classic *wg* LOF phenotype with interrupted wing margins (Fig. 4d). We tested the effects of each of the two single tgRNAs and found that the repression activity of the pair resided with *tgRNA-wg2*, which targeted a site at +28 bp relative to the TSS of the *wg* gene promoter (Fig. 4d). This wing margin-loss phenotype produced by *tgRNAs-wg(1 + 2)* or *tgRNA-wg2* was yet more pronounced when the *Cas9* was driven by the strong wing-specific *nubbin-Gal4* (*nub-Gal4*) source (Fig. 4e). Taken together, these results indicate that tgRNAs can efficiently repress the expression of both TATA-bearing and TATA-less target genes when targeting the TATA box (if applicable) or sequences near the TSS of these tested targets.

## Phenotypes generated by in vivo tgCRISPR activation and repression systems

Previous studies have shown that the *Cas9-VPR* fusion protein, when expressed with a *tgRNA*, can efficiently activate transcription in yeast or mammalian cells[16,17,19]. We tested whether the co-expression of Cas9-VPR with tgRNAs could similarly activate target gene expression during *Drosophila* development in a tissue-specific manner. We used *pannier-Gal4* (*pnr-Gal4*) to drive the expression of *Cas9* or *dCas9-VPR* in a broad dorsal stripe along the body axis[30]. In these experiments, lateral regions of the body serve as internal negative controls for tgRNA activity. We crossed *pnr-Gal4 > UAS:Cas9* or *pnr-Gal4 > UAS:dCas9-VPR* flies with animals carrying one or two tgRNAs targeting the *e* locus (*tgRNA-e1*, *tgRNA-e2*, or *tgRNAs-e(1 + 2)*) and examined body pigmentation phenotypes. As expected, *pnr-Gal4 > UAS:Cas9*, *tgRNAs-e(1 + 2)* as well as *pnr-Gal4 > UAS:Cas9*, *tgRNA-e2* individuals exhibited a LOF ebony phenotype in central but not lateral regions of the body wall indicating that efficient activity of tgCRISPRi was largely restricted to the zone of Cas9 expression (Fig. 5a). When we crossed the same *tgRNA* lines with those carrying the activating *pnr-Gal4 > UAS:dCas9-VPR* transgenes, we observed an opposite gain-of-function (GOF) phenotype with both *tgRNAs-e(1 + 2)*,

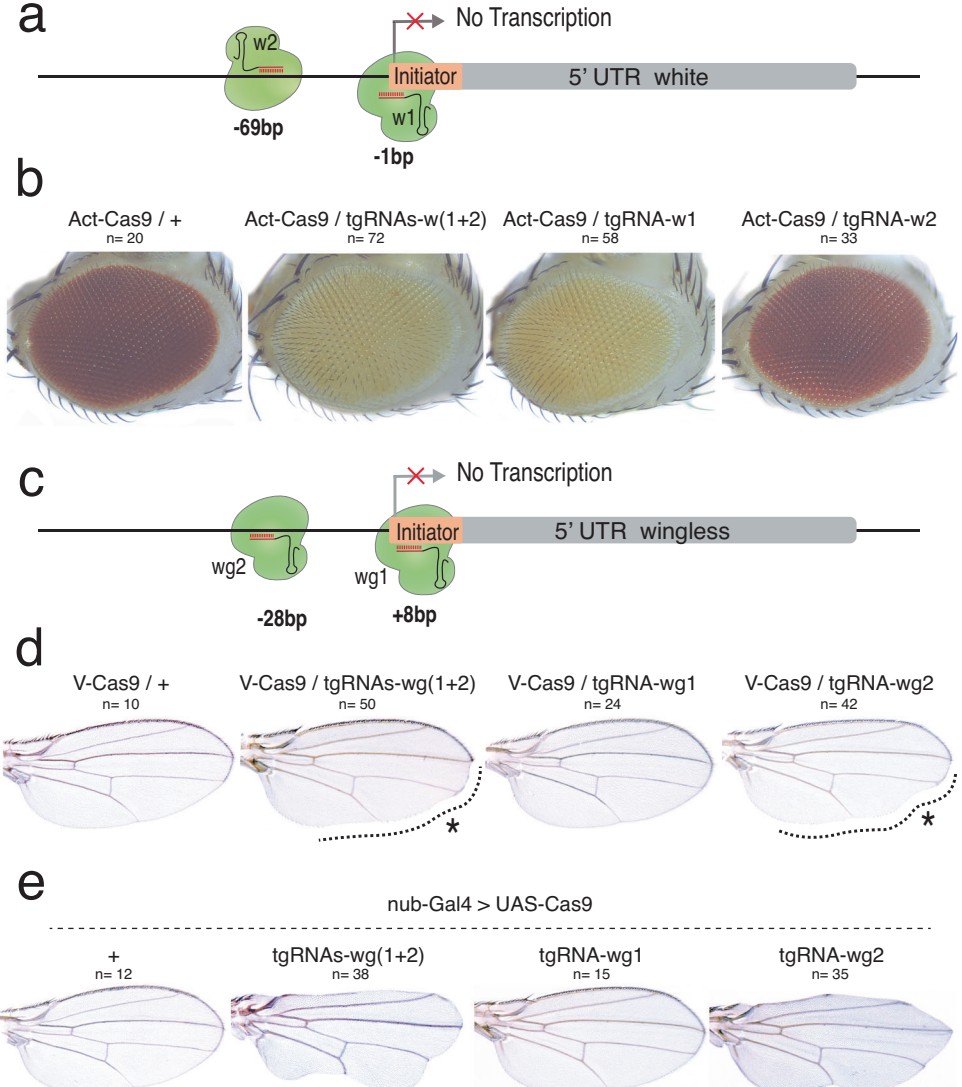

**Fig. 4 | Efficient repression of *w* and *wg* by tgCRISPRi. a** Genomic representation of the *w* locus showing the position of the sites targeted by the tgRNAs relative to the TSS. The coordinates of the tgRNA positions are indicated relative to the TSS. **b** Targeting the *w* gene using *Act-Cas9* and *tgRNAs-w(1+2)* and *tgRNA-w1* results in loss of eye pigmentation in flies. **c** Genomic representation of the *wg* locus showing the position of the tgRNAs designed for binding either upstream or downstream of TSS. The coordinates of the tgRNA positions are indicated relative to the TSS. **d** Targeting the *wg* gene using *vasa-Cas9* and *tgRNAs-wg(1 + 2)* and *tgRNA-w2* results in defects in fly wings. **e** Flies expressing Cas9 in the wing pouch by *nubbin-Gal4* (*nub-Gal4*) were crossed to males carrying the *tgRNAs-wg(1 + 2)*, *tgRNA-wg1*, and *tgRNA-wg2* construct, resulting in loss of wing margin tissue. *n* = number of samples.

and *tgRNA-e2*. This phenotype consisted of localized loss of pigmentation in regions of the dorsal thorax and abdomen, a previously documented phenotype resulting from overexpression of *e* (Fig. 5b)[38,39]. We analyzed ectopic expression and repression of the *e* target gene by RT–qPCR experiments in adult flies. In the case of ectopic expression in *tgRNA-e2/pnr-Gal4 > UAS:dCas9-VPR* flies, we observed an increase in *e* transcript levels (Sup. Fig. 5c). However, a significant decrease in overall *e* transcripts was not detected in *tgRNA-e2/pnr-Gal4 > UAS:Cas9* individuals compared to control flies, which presumably is due to the restricted pattern of *pnr-Gal4* expression since ubiquitous deployment of this same tgRNA with the more broadly active vasa-Cas9 source did reduce *e* transcript levels significantly (Fig. 2e, Sup. Fig. 5a). These penetrant and robust LOF and GOF phenotypes indicate that the same tgRNA(s) can sustain either repression or activation of a target gene when combined with Cas9 or Cas9-VPR, respectively.

## Employing tgCRISPR for enhancer-specific repression or activation of essential target genes

Previous studies have shown that CRISPRa or CRISPRi can be employed to target enhancers and cause epigenetic reprogramming by interfering with transcription factor binding, thereby leading to enhancer-specific modulation of gene activity[40–42]. These precise CRISPR targeting systems provide valuable opportunities for investigating the function of enhancers and other cis-regulatory modules (CRMs) in different developmental stages and diseases. We thus wondered whether the tgCRISPRi method could similarly be applied to alter the expression of an essential target gene in *Drosophila* by interfering with known transcription factor binding sites in CRMs controlling gene expression in a non-vital tissue. We designed tgRNAs targeting two such CRMs driving localized expression of the *Sex combs reduced* (*Scr* T1 CRM) and *knirps* (*kni* L2 CRM) genes (note that null alleles of both *Scr* and *kni* are homozygous lethal).

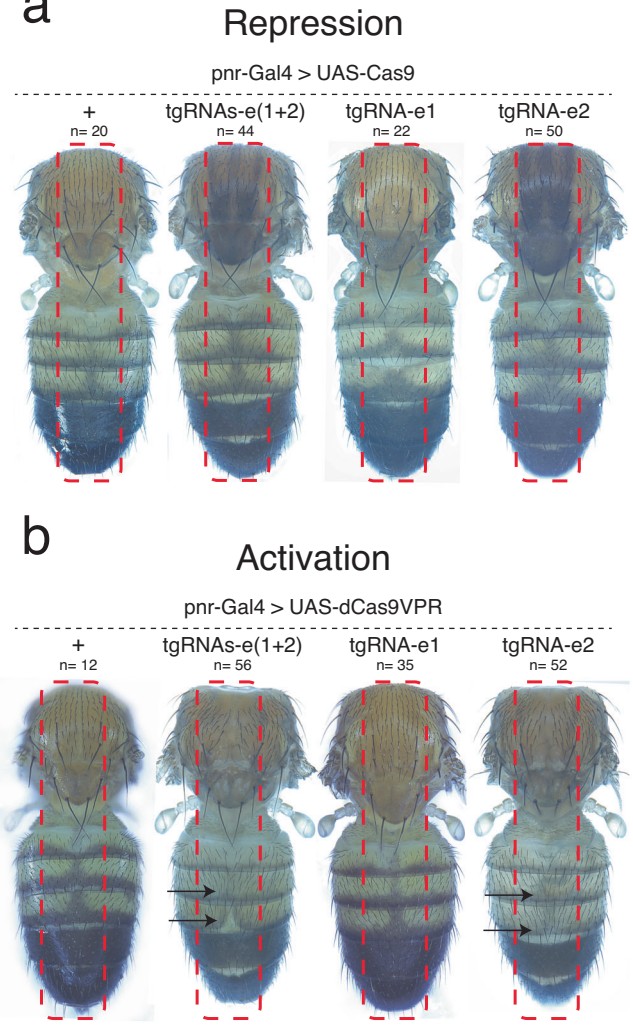

## a Repression

pnr-Gal4 > UAS-Cas9

+ n= 20  tgRNAs-e(1+2) n= 44  tgRNA-e1 n= 22  tgRNA-e2 n= 50

## b Activation

pnr-Gal4 > UAS-dCas9VPR

+ n= 12  tgRNAs-e(1+2) n= 56  tgRNA-e1 n= 35  tgRNA-e2 n= 52

**Fig. 5 | tgRNAs can sustain both repression (CRISPRi) and ectopic activation (CRISPRa) of the *e* locus. a, b** Flies expressing Cas9 (**a**) or dCas9-VPR (**b**) in the notum and dorsal cuticle by *pannier-Gal4* (*pnr-Gal4*) were crossed to males carrying the *tgRNAs-e(1 + 2)* construct. F1 progeny were screened for ectopic repression or activation of *e* locus phenotypes. The dotted rectangle indicates *pnr-Gal4* expression pattern. n = number of samples.

We first tested the inhibitory activity of tgRNAs targeting the T1 enhancer that drives *Scr* expression in defined domains of the first thoracic leg (T1 leg), which is required for the development of T1-specific sex combs in males[43,44]. Several Dll and En binding sites have been identified in the T1 CRM for *Scr* expression in T1 leg[43]. We designed a pair of *tgRNAs-Scr(1 + 2)* targeting Dll (*tgRNA-Scr1*) and En (*tgRNA-Scr2*) binding sites (Fig. 6a). We crossed *Act5C-Cas9* flies with animals carrying these two tgRNAs and observed that F$_1$ males expressing both Cas9 and the tgRNAs had a markedly reduced number of T1 sex comb bristles (average = 8 ± 2) (Fig. 6b, c) compared to either wild-type males or *Act5C*-Cas9-only controls (average = 11 ± 2) (Fig. 6b, c), thus mimicking phenotypes known to be associated with reduced *Scr* expression or activity[43,45].

We also tested whether tgRNAs could regulate gene expression via enhancer binding in the context of a *knirps* (*kni*) CRM which is required for restricted *kni* expression in the L2 wing vein primordium of third larval instar wing discs[46,47]. Several Aristaless (Al), Scalloped (Sd), Engrailed (En), Optix (Opt), and Brinker (Brk) transcription factors binding sites have been shown to play important roles in restricting *kni* expression to the L2 vein primordium[46,48]. We created two tgRNAs

(*tgRNAs-kni(1 + 2)*) which target L2-CRM, with one located near the Al binding site (*tgRNAs-kni1*) and the other located near the Opt binding site (*tgRNAs-kni2*) (Fig. 6d). We tested the activity of this tgRNA pair by crossing flies carrying *tgRNAs-kni(1 + 2)* to strains expressing either wild-type Cas9 or Cas9-VPR and examined wing phenotypes in F$_1$ adults. When flies carrying *tgRNAs-kni(1 + 2)* were crossed to wild-type Cas9 producing strains, we observed no obvious wing phenotypes in F1 progeny carrying a single copy of both transgenes (Fig. 6e). However, in flies carrying two copies of each *tgRNAs-kni(1 + 2)* and Cas9, we noticed a significant anterior shift in the position of the L2 vein (Fig. 6f, g; Sup. Fig. 6). The L2 vein was also shortened as a consequence of its being shifted anteriorly compared to the control (Act-Cas9/Act-Cas9) (Fig. 6g), indicating Cas9/tgRNAs interfering with L2-CRM activity. In a previous study, we observed a similar phenotype when a subdomain of the L2-CRM containing Optix bind sites, which we refer to as the "repression domain", was deleted, resulting in a comparable anterior shift of the L2 vein[47]. These observations suggest that tgRNA-mediated binding of active Cas9 to key sites within the L2-CRM repression domain interferes with the action of transcriptional repressors that would otherwise bind to such sites to exert a repressive function.

We also tested whether *tgRNAs-kni(1 + 2)* could mediate wing-specific activation of *kni* when combined with either the *vasa-Cas9-VPR* (fused to wild-type Cas9) or *1096-GAL4/UAS-dCas9-VPR* sources. Indeed, we observed wing-specific defects in such individuals that were characteristic of *kni* overexpression (Fig. 6e), as previously described[46]. These ectopic wing vein phenotypes were similar to those produced by *kni* mis/overexpression, consistent with the possibility that they reflect gain-of-function activity of the cis-acting L2-CRM. We tested this hypothesis by making use of a strain of flies in which the L2-CRM from *D. melanogaster* has been swapped with that from a distantly related species within the Drosophilid group (*Drosophila grimshawi*). In a previous study, we showed that the *D. grim.* L2-CRM (L2-GrimCRM) could substitute for its *D. mel.* counterpart to produce a wild-type L2 vein pattern[47]. Importantly, for our current purposes, the *D. grim.* and *D. mel.* L2-CRMs share very little primary sequence similarity. In particular, *tgRNAs-kni(1 + 2)* should not be able to bind to sequences in the L2-GrimCRM where they are absent. As expected, flies carrying two copies of the L2-GrimCRM construct carrying *tgRNAs-kni(1 + 2)* and the *vasa-Cas9-VPR* source had normal wings and L2 vein patterns (Fig. 6i). However, when these flies were crossed with wild-type *D. mel.*, which carries an endogenous L2-CRM, similar wing vein defects were observed as described above (Fig. 6i). These CRM-swap experiments demonstrate that the wing defects we observe are mediated by *tgRNAs-kni(1 + 2)* binding to endogenous L2-CRM sequences and are not due to some indirect effect of the tgRNAs acting on some other targets influencing the L2 vein gene-regulatory network.

### Concurrent gene-drive (gRNA) and transcriptional repression (tgRNA) using active Cas9

One potential use of tgCRISPR technology is in the field of gene-drives, wherein a full-length gRNA instructs copying of the gene drive-element, while one or more tgRNAs could modulate the expression of other genes (for example, to promote HDR-based copying events over creation of NHEJ mutant alleles, or to add functionalities to the gene drive element). We designed a proof-of-principle experiment, in which a gRNA targeting cleavage at *y* permits copying of a gRNA split-drive cassette (y-Drive) onto a homologous receiver chromosome, while an additional tgRNA-e1 simultaneously represses somatic expression of the *e* gene in the presence of active Cas9 (Fig. 7a).

We tested the efficiency of the y-drive by producing F$_1$ transheterozygous females carrying *vasa-Cas9* and the *gRNA-y* cassette (y-Drive) (Fig. 7a). These flies displayed the expected mosaic *yellow* mutant pigmentation phenotype with patches of *y$^+$* tissue (Fig. 7b). Germline inheritance of the drive cassette in such F$_1$ females was assayed in individual crosses with wild-type males (Fig. 7a). Phenotypic

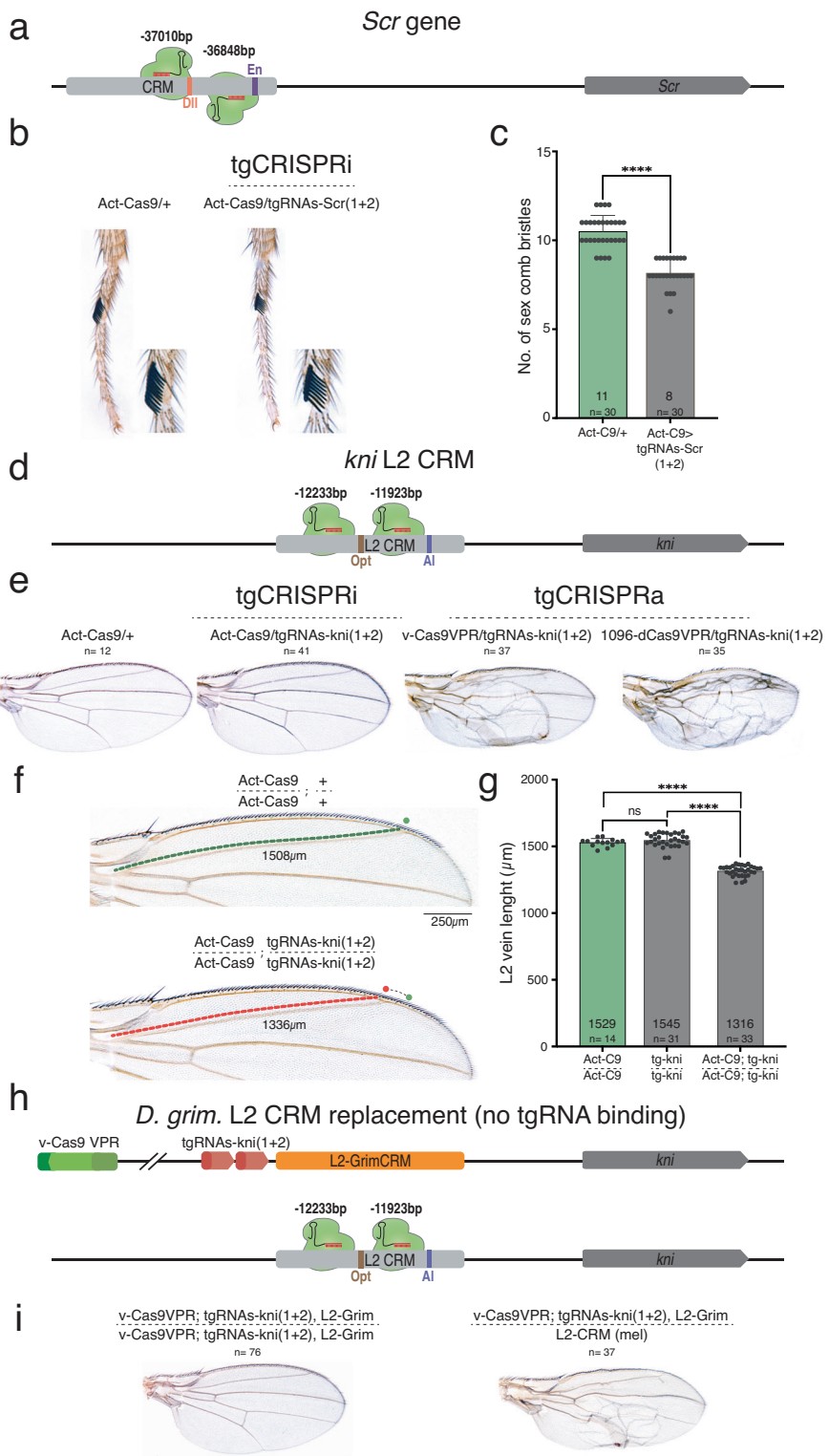

analysis of the fluorescent markers in the resulting $F_2$ progeny allowed simultaneous evaluation of the germline inheritance rates of *gRNA* copied transgene (y-Drive) and a vasa-Cas9 source with or without the tgRNA targeting the *e* locus (Fig. 7c). In the absence of the tgRNA component, we found efficient Cas9-driven super-Mendelian inheritance of the y-drive transgene (average = 82%) as previously observed. Next, we evaluated the effect of the *tgRNA-e2* (provided from the 3rd chromosome with the Cas9-expressing transgenic cassette) transgene on transmission of the y-drive (Fig. 7a). $F_1$ trans-heterozygous females carrying all 3 constructs displayed the expected full-body *yellow* and

*ebony* mutant pigmentation phenotypes (Fig. 7b). In $F_2$ progeny, we observed comparable inheritance rates for the y-Drive transgene (average = 81%) with concurrent transcriptional inhibition of *ebony* gene expression, indicating that inclusion of tgRNA-e2 did not alter drive activity of the full-length gRNA-y (Fig. 7c).

### Repression of a transcriptional reporter using tgRNAs in mammalian cells

Previous studies have shown that tgRNAs can direct a Cas9-VPR fusion protein to bind target sequences without DNA cleavage. These studies

**Fig. 6 | Enhancer-specific transcription modulation of *Scr* T1-CRM and *kni* L2-CRM by tgCRISPR system. a** Genomic representation of the *Scr* locus with T1-CRM showing the position of the sites targeted by the tgRNAs on T1-CRM. The coordinates of the tgRNA positions are indicated relative to the TSS. **b** Targeting the T1-CRM of *Scr* gene using *Act-Cas9* and *tgRNA s-Scr(1 + 2)* results in reduced number of bristles in sex comb in male first thoracic legs. **c** Graph shows comparison of number of bristles in sex comb in male first thoracic legs in control (*Act-Cas9/+*), and F1 male progeny generated by either tgRNA with *Act-Cas9* flies (n = 30 male T1 legs). Data were analyzed using a two-tailed unpaired t-Test. Data plotted as mean ± standard deviation, \*\*\*\*p < 0.0001. **d** Genomic representation of the *kni* locus L2-CRM showing the position of the tgRNAs targeting L2-CRM. The coordinates of the tgRNA positions and are indicated relative to the TSS. **e** Targeting the L2-CRM of *kni* gene using *tgRNAs-kni(1 + 2)* and *Act-Cas9* resulted in no noticeable phenotype in fly wings. Flies expressing either *vasa-Cas9-VPR* or Cas9-VPR by *1096-*

*Gal4* (*1096-Gal4*) crossed to males carrying the *tgRNA s-kni(1 + 2)*, results in blister-like defects in fly wings. **f** Fly wing L2 vein phenotype from homozygous *Act-Cas9* (L2 vein length indicated by a green dotted line) and from homozygous *tgRNAs-kni(1 + 2)* and *Act-Cas9* (L2 vein length indicated by a red dotted line). An anterior shift of L2 vein is indicated with green (wild-type) versus red (shifted) dots. **g** Graph shows the comparison of L2 vein length in control flies homozygous for *Act-Cas9* and *tgRNAs-kni(1 + 2)*, with flies homozygous for *Act-Cas9; tgRNAs-kni(1 + 2)* flies. Data were analyzed using One-way ANOVA. Data plotted as mean ± standard deviation, \*\*\*\*p < 0.0001, ns = not significant. **h** Genomic representation of the *kni* locus L2-CRM replaced with L2-GrimCRM and *D. mel.* wild-type L2-CRM showing the tgRNAs targeting L2-CRM. **i** Fly wing phenotype from homozygous L2-GrimCRM and from heterozygous L2-GrimCRM/L2-CRM with *tgRNAs-kni(1 + 2)* and *vasa-Cas9-VPR*. n = number of samples.

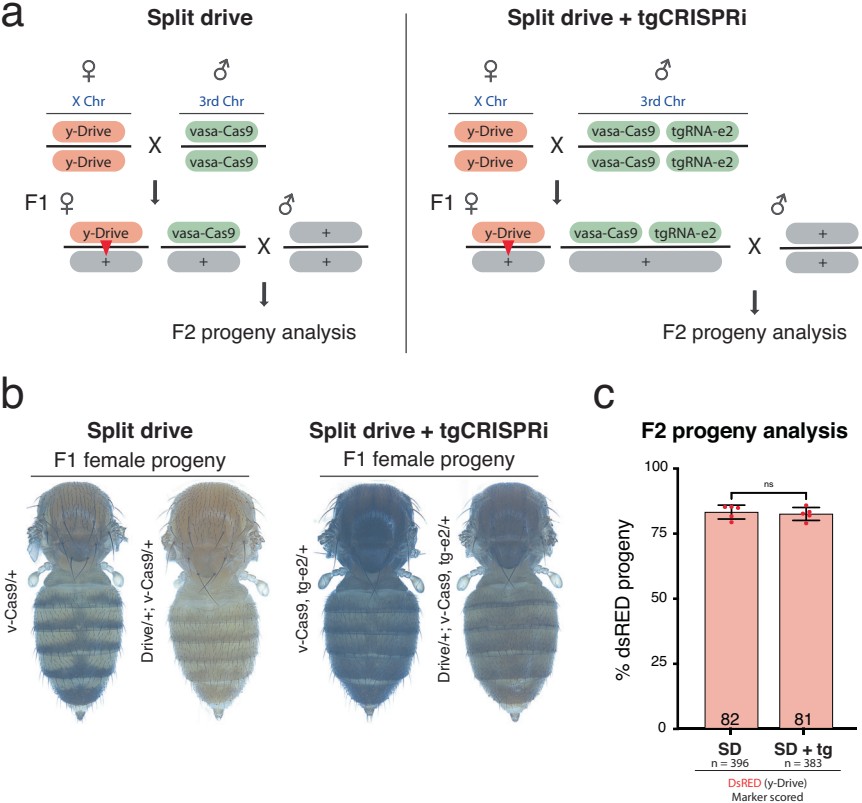

**Fig. 7 | Concurrent drive (via a gRNA) and transcriptional repression (via a tgRNA) using active Cas9. a** Outline of the genetic cross used to test split-drive (SD) and SD + tgCRISPRi in fruit flies, indicating transgene and wild-type allele locations on different chromosomes. The sex of the individuals is indicated with the symbols "♂" for males and "♀" for females. $F_0$ females carrying a DsRed-marked gRNA-y transgene (y-Drive) inserted in the *yellow* locus were crossed with males carrying a GFP-marked cassettes containing Cas9 and Cas9+tgRNA-*e2* on 3rd chromosome. Transheterozygous $F_1$ females (carrying Cas9 + gRNA and +/− tgRNA) were crossed with wild-type males to assess germline transmission rates of fluorescently marked transgenes and tgCRISPRi LOF phenotypes in the $F_2$ progeny. Conversion events are indicated by red triangles in the $F_1$ females. **b** $F_1$ female progeny body pigmentation phenotype from SD only (*y⁻* phenotype) and SD +tgCRISPRi (*y⁻* and *e⁻* phenotypes). **c** $F_1$ female germline inheritance output was measured as DsRed marker presence in the $F_2$ progeny. The bar represents the inheritance average of the DsRed-marked drive cassette with or without tgCRISPRi. Inheritance averages are represented in a bar graph with the respective data. Data were analyzed using two-tailed unpaired t-Test. Data plotted as mean ± standard deviation, ns = not significant, *n* = number of samples.

also revealed that expression of a Cas9-VPR fusion protein with a tgRNA could lead either to transcriptional activation or repression of target genes in mammalian cells depending on the context[16]. Based on these observations and the findings described above in *Drosophila*, we hypothesized that targeting complexes of active unmodified Cas9 and tgRNAs to sites near the TSS could also repress target gene expression in mammalian cells.

We tested this hypothesis using a set of tgRNAs targeting the CMV promoter driving the expression of an mCherry reporter gene in transiently transfected human HEK293T cells (Fig. 8a). Based on our

initial observations, we designed two tgRNAs of 15 nt targeted to positions −33 (*tgRNA-C2*, close to TATA box) and at +1 (*tgRNA-C1*, at TSS initiator sequence) of the CMV promoter TSS (Fig. 8a). A plasmid encoding both Cas9 and the dual tgRNAs was transfected along with the CMV-mCherry reporter plasmid into HEK293T cells. After 72 h, cells were harvested and analyzed using flow-activated cell sorting (FACS) (Fig. 8b, Sup. Fig. 7). We observed ~40% reduction in frequency of mCherry-expressing cells with *Cas9/tgRNA-C2*, ~60% reduction with *Cas9/tgRNA-C1*, and ~70% reduction with both *Cas9/tgRNAs-C(1 + 2)* (Fig. 8c, d). Thus, in accordance with our results from *Drosophila*, the

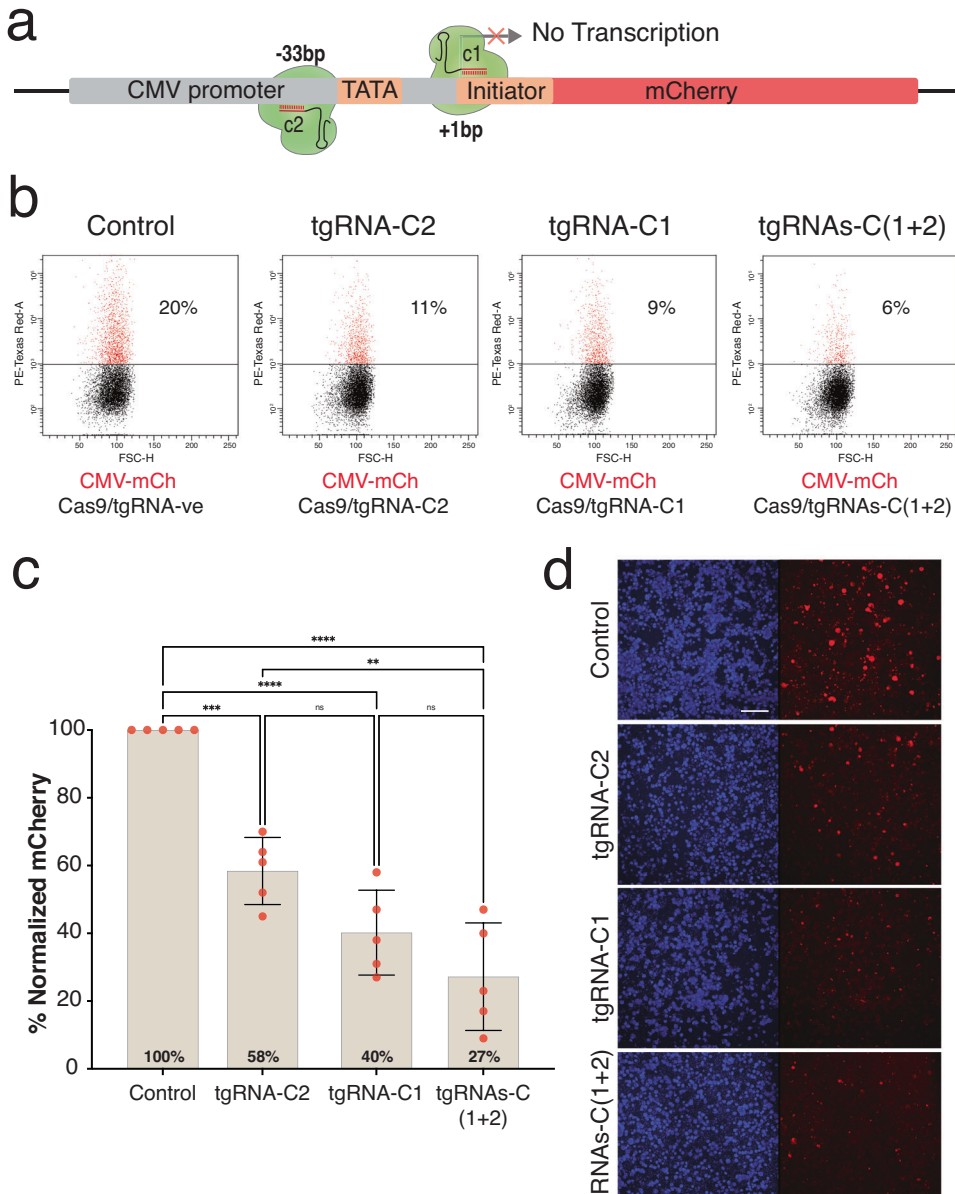

**Fig. 8 | Repression of a transcriptional reporter using tgRNAs and active Cas9 in mammalian cells. a** A schematic of a reporter construct driven by the CMV promotor showing the positions of the tgRNA designed either upstream or downstream of TSS. The coordinates of the tgRNA positions are indicated relative to the TSS. **b** Pseudocolor FACS plots of analyses of mCherry fluorescence in HEK293 cells transfected with CMV-mCherry plus Cas9-tgRNA-ve plasmid (control) or CMV-mCherry plus Cas9-*tgRNA-c*2 plasmid (*tgRNA-c2*), or CMV-mCherry plus Cas9-tgRNA-c1 plasmid (tgRNA-c1), or CMV-mCherry plus Cas9-*tgRNAs-c(1 + 2)* plasmid (*tgRNAs-c(1 + 2)*). mCherry-positive cells are shown in red and representative percentages (%) are indicated. **c** Normalized % of mCherry-positive cells from FACS data was plotted. Five independent experiments were performed and analyzed using One-way ANOVA. Data plotted as mean ± standard deviation, **$p = 0.0024$, ***$p = 0.0001$, ****$p < 0.0001$, ns = not significant. **d** Images of HEK293T cells 72 h after transfection with CMV-mCherry construct and Cas9-tgRNAs targeting CMV promoter (mCherry in red; DAPI in blue; scale bar = 100 μm).

tgCRISPRi system when targeted close to a TSS could significantly repress reporter gene expression using a standard active form of Cas9 in mammalian cells. In this instance, both tgRNAs contributed in a collaborative fashion to repression of target gene expression.

## Discussion

In this study, we demonstrate in vivo efficacy of a tgCRISPRi system that employs a catalytically active form of Cas9 in combination with truncated guide RNAs to knock-down endogenous gene expression of several targeted loci in *Drosophila*. The LOF phenotypes observed with tgRNAs result from repression of target gene transcription associated with Cas9 binding near the TSS without any detectable DSBs generated

at the target sites. Furthermore, beyond the expected LOF phenotypes observed with the various tgRNAs tested, we found no evidence of any undesired off-target phenotypes. Analysis of several tgRNAs performing individually to efficiently repress target gene expression suggests that the ideal target region includes the TATA box and the initiator region of the TSS. We found that single tgRNAs targeting the TATA box or the TSS can fully repress target gene expression. Given that not all tgRNAs are efficient, a good general strategy for optimizing the likelihood of success would be to express two sgRNAs per target gene from a single plasmid such as the convenient vector pCFD4[32]. Future studies applying tgCRISPR to an extended set of gene targets may reveal more precise sequence or positional parameters to enable

more accurate predictions or rules for how to design efficient tgRNAs. The relative locations of nucleosomes or other chromatin modifications may also be relevant in this context which could reduce binding of gRNAs that would otherwise bind well to naked DNA, effects which might be mitigated by potential modifications to the Cas9 protein.

Our results also demonstrate that the tgCRISPR system can be used both for in vivo repression and activation of target genes in *Drosophila*. Introducing Cas9 or Cas9-VPR along with tgRNAs allows targeted gene repression or activation in a tissue-specific manner. This precision in gene modulation can be further refined by using tgRNAs that target sites in specific CRMs of essential genes, although the general rules for choosing such targets will need to be examined further. The ability of a single active Cas9 protein to regulate transcription while maintaining the capacity to cleave DNA to edit specific gene targets or catalyze localized gene conversion events expands the active genetics toolbox and should aid in deciphering complex biological interactions. The Perrimon group has developed a genome-scale collection of CRISPRa transgenic gRNA lines for overexpression screens in *Drosophila*[11]. Similar to this system, the tgRNA strategy that we describe herein should permit the development of targeted sets of genes for knock-down and overexpression. Also, if rules can be inferred for which tgRNAs are most likely to be effective, tgCRISPRi could potentially be exploited for genome-scale transgenic collections, thus complementing other approaches such as UAS-RNAi lines[49] and CRISPRa systems[11,14,15]. While use of genome-wide GAL4-driven UAS-RNAi lines have proven invaluable for research in *Drosophila*, we note that such comprehensive resources do not currently exist for other systems such as mosquitoes. In such scenarios, tgCRISPRi may offer a particularly valuable alternative for gene suppression. Also, RNAi typically requires fairly high levels of expression that are readily generated by the GAL4-UAS system, which would be bulky to include as an auxiliary element. In some contexts, however, RNAi constructs could be effective when expressed directly by a pol-II promoter. A question to address in the future is whether tgCRISPR will prove to have fewer off-target effects than those that occur with RNAi, which in some contexts can pose significant challenges.

One attractive application of tgRNA-based gene silencing is the potential amelioration of gene-drive systems. In proof-of-principle experiments, we combined a split-drive element with a tgRNA and an active vasa-Cas9 source. In these experiments, we observed both efficient drive of the y-Drive component inserted into the y locus (81%) as well as complete (100%) silencing of the *e* gene targeted by the tgRNA. We obtained a similar inheritance rate of the y-Drive transgene with or without concomitant transcriptional suppression of *e* gene expression. These results demonstrate that tgRNAs could be harnessed for augmenting the functionality of gene-drives with the simple addition of compact tgRNA components. The observation that the gene-conversion efficiency mediated by the drive gRNA is not diminished by co-expression with tgRNA (e.g., through competition for a potentially limiting amount Cas9) is encouraging in this regard, and is consistent with other reports indicating that Cas9 can maintain its driving activity in the presence of multiple gRNAs that do not participate in the drive process[50]. One example of how gene-drive systems might benefit from additional tgRNAs is to employ them for suppressing unproductive and disruptive NHEJ events, or alternatively, using such tgRNAs in an activating mode to stimulate the desired HDR pathway to enhance drive efficiencies. In addition, tgRNAs could be included to modulate the activity of second-site loci mediating insecticide resistance or parasite transmission. As an example of the latter use, including a tgRNA to suppress expression of the FREP1 gene in developing gut cells of Anopheline mosquitoes could help reduce transmission of malarial parasites since the FREP1 protein plays an important role in these parasites transiting the gut epithelium into the body cavity to gain access to salivary glands[51]. Furthermore, Cas9-VPR and tgRNA components could be employed to overexpress immune

pathway genes in mosquitos to minimize pathogen survival[52]. Multiple factors need to be considered when developing tgCRISPR-augmented gene-drive systems. For instance, modifying the expression of a second site gene in a somatic tissue like the midgut using a tgRNA could result in expression and activity of the full-length gRNA used for the drive in that tissue. This may not be desirable if the goal is to optimize drive efficiency by limiting Cas9 expression to the germline. One possible solution is to use a pol-II-promoter to express the driving gRNA as well as Cas9 in a germline-specific pattern, using self-splicing gRNA constructs such as hammerhead or tRNA scaffolds. In such a scenario, the tgRNA could be expressed using a standard ubiquitous pol-III promoter or in a tissue-specific pattern matching the targeted cell type. Another consideration is that if the tgRNA produces a fitness cost, it could lead to the evolution of target site mutations rendering it inactive when combined with a drive. This issue might be addressed by using two or more tgRNAs that efficiently act on a given target gene to provide redundancy in target gene modulation.

Finally, we also note that the CRM-swap strategy described in this study to demonstrate that the *tgRNAs-kni(1 + 2)* act to mis-regulate the *kni* gene via its cis-acting L2-CRM could also be used in principle to replace an entire population of a wild species with an alternate CRM lacking tgRNA binding sites without creating any DNA damage (e.g., NHEJ induced indels), which would be associated with a drive system inserted into this locus. Such systems could function in a manner akin to the *ClvR* strategy[50]. Thus, the ability to employ tgRNAs for targeting both promoter-proximal and CRM regulatory regions provides a variety of options for designing population replacement strategies.

tgCRISPR-based strategies might also be employed in the field of gene therapy. For example, overexpression of endogenous homology-based repair factors such as Rad51 could help shift the response to Cas9-induced DSBs from mutagenic NHEJ to precise interhomolog repair (HTR), in which a donor plasmid or the homologous chromosome are used as a template to repair a cleaved allele[53]. Similarly, tgCRISPR systems could be used to improve Cas9-nickase-mediated allelic HTR repair[54] for correcting dominant or trans-heterozygous disease-causing DNA alterations. Previously we have shown that Cas9-nickase, which nicks rather than cleaves target DNA, induces few NHEJ events and sustains higher rates of somatic allelic conversion than those observed with Cas9[54]. Thus, tgRNAs could be used in combination with allele-specific full-length gRNAs and Cas9-nickase to knock-down (or with Cas9-nickase-VPR to overexpress) critical factors that improve allelic repair without creating any unwanted genome damage. In aggregate, these studies may guide the design of strategies in which a single active enzyme nuclease (Cas9 or Cas9-VPR) combined with a set of gRNA/tgRNAs can cleave DNA for gene editing at one locus while modulating expression of other genes for optimized outcomes. Such strategies should expand possibilities for a wide range of CRISPR applications.

## Methods

### *Drosophila* rearing and fly stocks
Fly stocks were maintained at 18 °C, whereas crosses were carried out at 25 °C on a 12-h day/night cycle on corn-meal media. vasa-Cas9-GFP (BDSC #51324), Act-Cas9 (BDSC #54590), nubGal4>UAS-Cas9 (BDSC #67086), nubGal4>UAS-dCas9-VPR (BDSC #67055), 1096Gal4>UAS-dCas9-VPR (BDSC #67039), pnrGal4>UAS-Cas9 (BDSC #67077) and pnrGal4>UAS-dCas9VPR (BDSC #67040) were procured from Bloomington *Drosophila* Stock Center (BDSC). Other lines used in the study were generated during this study.

### Microinjection of tg/gRNA constructs
Gene-specific tgRNAs and gRNAs were designed using an online gRNA tool (flycrispr) and inserted into a previously described compatible gRNA scaffold construct[55]. Single tg/gRNA constructs were cloned in the pCFD3 vector, and tandem tg/gRNAs were cloned in the pCFD4

vector[55]. gRNA and tgRNA sequences are listed in Supplementary Table 1. Primers used for cloning are listed in Supplementary Table 2. Plasmids were prepared using the Qiagen Plasmid Midi Kit (#12191), and their sequence was confirmed. All tg/gRNA and vasa-Cas9-VPR constructs were injected at the attp2 landing site on the third chromosome (BDSC #25710 and 8622) using a service offered by Rainbow Transgenic Flies, Inc. Vermillion or GFP positive males were selected in the F$_1$ progeny to establish isogenic lines and specific insertions were confirmed by sequencing.

### Sample preparation for sequencing
For Sanger sequencing, genomic DNA was extracted from a single fly per replicate following the method described by Gloor GB and colleagues[56]. tgRNA and gRNA target sites were amplified with the help of specific primers and using the Q5 Hot-start master mix (NEB #M0494S). PCR products were purified prior to Sanger sequencing. Data were analyzed using an ICE tool by Synthego (https://ice.synthego.com/).

### Mammalian cell culture and transfection
HEK293T cells (ATCC, Cat No: CRL-3216) were maintained in high glucose Dulbecco's modified Eagle's medium (Invitrogen) supplemented with 10% fetal bovine serum (Invitrogen) and 1% penicillin/streptomycin (Invitrogen). The cells were maintained at 37 °C and 5% CO$_2$ in a humidified incubator. The cells were seeded in 24-well plates at a density of 50,000 cells/well for transfection. In total, 1000 ng of Cas9-tgRNA and 500 ng of reporter plasmid (when required), along with Lipofectamine 2000 (#11668019) were added to each well as per the manufacturer's instructions. The cells were grown for 72 h after transfection and subsequently analyzed using fluorescence microscopy and FACS.

### Cell staining
The cells were grown on polylysine-coated coverslips and fixed in 4% PFA for immunocytochemistry. The fixed cells were washed three times with PBS-T for ten minutes each. Subsequently, the fixed cells were stained using DAPI (1:1000 in PBS) for 10 min at room temperature. Finally, the fixed cells were washed three times with PBS for 5 min each and mounted on a microscope slide with PVA DABCO (Sigma-Aldrich).

### Microscopy
Adult flies were anesthetized using CO$_2$, and z-stack images were acquired on a Zeiss fluorescence microscope. Helicon Focus software was used to stack the images.

### Quantitative PCR assay
Adult flies with genotype *tgRNA-w/Act5C-Cas9* expressing tgRNAs for *white* and *tgRNA-e/Act5C-Cas9* expressing tgRNAs for *ebony* were collected. For tgCRISPRi and tgCRISPRi fly lines with the genotype *tgRNA-e/+;UAS:Cas9/pnr-GAL4* and *tgRNA-e/+;UAS:dCas9-VPR/pnr-GAL4* were used. The corresponding sibling flies obtained from the cross served as controls.

RNA was extracted and purified by Direct-zol RNA Kits (Zymo #R2051) following the manufacturer's protocol. The isolated RNA was used to synthesize cDNA, following the protocol for the High-Capacity cDNA Reverse Transcription Kit (Applied Biosystems). The RT-qPCR experiments were conducted using LightCycler 480 SYBR Green I Master (Roche) in a Roche Real-Time PCR LightCycler 96. The threshold cycle (C$_T$ or $2^{-\Delta\Delta CT}$) method[57] was used to calculate the fold-change for transcript relative quantification. Three independent biological replicates were analyzed in each case. Statistical analysis (Student's t-Test and one-way ANOVA test) were performed using the GraphPad Prism 9 software. Primers used for RT-qPCR experiments for each target are listed in Supplementary Table 2.

### Statistical analysis
GraphPad Prism 9 was used for all statistical analyses and graphical visualization of data.

### Public ChIP-seq data analysis
ChIP-seq data from S2-DRSC cells were previously published[58]. Raw data was downloaded from the GEO database GSE101557. Fastq files were aligned to the dm3 genome using Bowtie2 software and peak calling was performed using MACS2. BigWig files were visualized using the IGV software.

### Reporting summary
Further information on research design is available in the Nature Portfolio Reporting Summary linked to this article.

## Data availability
The sequence of all gRNAs, tgRNAs, and oligoes used in this study are provided with this paper in Supplementary Tables 1 and 2. RNA polymerase II subunit C (Rpb3) (GSE101557) ChIP-seq dataset was used in this study. Source data is provided in this paper as a Source data file. Source data are provided with this paper.

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

## Acknowledgements

We thank all members of the Bier laboratory for constructive ideas and discussions. These studies were supported by NIH grants R01GM117321, R01GM144608, R01AI162911, and The Bill and Melinda Gates Foundation (B&MGF) awarded to Ethan Bier.

## Author contributions

A.A.: Contributed to the original concept of tgCRISPR, designed and contributed to conducting experiments, and writing the manuscript. S.K.: Contributed to cell culture experiments, and editing the

manuscript. A.G.: Contributed to designing and editing the manuscript. M.S.: Contributed to fly experiments and editing the manuscript. E.B.: Contributed to experimental design, and writing the manuscript.

## Competing interests

E.B. has equity interest in Synbal, Inc. and Agragene, Inc., companies that may potentially benefit from the research results and also serves on the company's Board of Directors and Scientific Advisory Board. The terms of this arrangement have been reviewed and approved by the University of California, San Diego in accordance with its conflict of interest policies. The other authors declare no competing interests.
