## [Peer Review File · Nature Communications]

Reviewers' Comments:

Reviewer #1:

Remarks to the Author:

In this study Auradkar et al. use truncated sgRNAs (tgRNAs) in combination with wildtype Cas9 for transcriptional interference of target genes. Working mostly in *Drosophila*, the authors show that such a CRISPRi system can repress the expression of four endogenous target genes to sufficient levels to observe loss-of-function phenotypes. They also demonstrate that tgRNAs can be combined with a conventional nuclease driven gene drive and demonstrate inhibition of a transgenic reporter in HEK293T cells.

While the experiments described in this manuscript are sound and the data support most of the conclusions, I do not believe that this study presents a sufficient conceptual advance or a method of broad enough interest to warrant publication in *Nature Communications*. I will substantiate this view in the major points below.

Major points:

The tgCRISPRi system does not present a significant conceptual advance. The fact that truncated sgRNAs can mediate binding of Cas9 to the target locus without activating nuclease activity has been demonstrated on multiple occasions and that recruiting Cas9 to the target locus without cutting it can elicit a CRISPRi effect is equally well known. Moreover, previous studies have directly demonstrated that combining tgRNAs and Cas9 can be used to modulate transcription. Santos-Moreno et al. and Jeong et al. both have shown that truncated sgRNAs can mediate CRISPRi in prokaryotes (PMID: 36787424, PMID: 32488086). Furthermore, the very first description of CRISPRi in mammalian cells already tested truncated sgRNAs and showed that guides shortened by up to 8 nucleotides can mediate gene repression (PMID: 33545038). Maybe most significantly, Ye et al. have used tgRNAs and catalytically active Cas9, the exact method described here, for CRISPRi in mammalian cells (PMID: 30062046). In summary, iterations of the tgCRISPRi system have already been described on multiple occasions. The fact that none of the references listed above are currently cited in the manuscript suggests that the authors could improve the referencing of prior work.

The tgCRISPRi system is unlikely to be a method of widespread interest. This paper mainly describes the use of tgCRISPRi in *Drosophila*. CRISPRi is relatively under-explored in flies, almost certainly because the *Drosophila* community already has an excellent system for gene knockdown in the form of RNAi. There are several large-scale transgenic resources for RNAi in flies that are publicly available and together cover almost the entire genome. They are undoubtedly one of the most widely used tools in the *Drosophila* community. Although RNAi works very well in flies compared to other systems, it is not without its drawbacks. Most importantly, gene knockdown is typically incomplete and there is a significant fraction of RNAi lines where knockdown is so inefficient that no phenotype can be observed (PMID: 26320097). However, the tgCRISPRi system suffers from the exact same drawbacks. The knockdown efficiencies of the tgRNAs tested in this study are well within the range typically observed with RNAi (PMID: 26320097) and also include a significant proportion of inactive tgRNAs. Thus, this new method essentially does what current state-of-the-art RNAi does without improving on its limitations. Off-target effects are another limitation of RNAi, but the current manuscript does not test for on-target specificity of tgCRISPRi. For this reason, it is hard to see why many people would go through the trouble of generating new tgRNA transgenic lines when there are already RNAi lines that can be used for the same purpose. There is more utility for this system in mammalian cells, where RNAi is more problematic, but as outlined above, tgCRISPRi has already been described in that system.

At the beginning of the manuscript the authors compare the phenotypes caused by tgCRISPRi targeting yellow and ebony with those caused by Cas9-mediated indel mutagenesis using full-length sgRNAs targeting the same target sites. This comparison can be misleading, as sgRNAs for CRISPRi/a or CRISPRn should be designed differently. While guides for transcriptional interference should be targeted near the transcriptional start site (as done here), sgRNAs for gene disruption

with CRISPR nucleases should be targeted to the coding sequence, where out-of-frame indels are likely to disrupt gene function. Therefore, the experiments in this manuscript compare correctly designed tgCRISPRi guides to incorrectly designed CRISPRn guides. I see the point of doing these experiments to demonstrate the effect at the on-target sites at sequence level, but with regard to phenotypic penetrance, readers would need to be alerted in the main text that this constitutes comparing apples with oranges. Furthermore, the statement in lines 87-89 should be removed for the same reason.

The amount of knock-down necessary to generate a loss-of-function phenotype is highly gene specific (e.g. ebony is known to be partially haploinsufficient). In contrast, quantification of transcript abundance is a gene agnostic measure on the efficiency of tgCRISPRi. It would strengthen the manuscript if it were performed on all targets.

Minor points:

Consider changing CRISPR/Cas to CRISPR-Cas.

Line 132: The target sites are not in the reading frame.

Line 203: - 129 not + 129

Line 340: As outlined above the tgCRISPRi system has been described before.

Line 350 and line 374: remove "fully" and "complete (100%)". All shown tgRNAs lead to knock-downs of target gene expression of no more than 80%.

Figure 5b: The pigmentation phenotype is hard to see, but there are clear signs of toxicity (fusion defects in thorax and abdomen). Please discuss this in the results section.

Reviewer #2:

Remarks to the Author:

This is an interesting manuscript in which the authors explore what can be done with truncated gRNAs alone, or in combination with long gRNAs, all in the context of Cas9.

As background, it has been shown earlier that long gRNAs (called just gRNAs from here out) bound to dead Cas9 can inhibit transcription when bound near a transcription start site. It has also been shown that truncated gRNAs (tgRNAs) bound to active Cas9 cannot activate cleavage but can, in the context of a Cas9-VPR fusion protein, activate transcription (though this has not, until this paper, been shown in *Drosophila*). In this latter context (mammalian cells) it has also been shown that this fusion can do two things at once, cleave one set of sites and activate transcription from others.

What has not been shown (I think) is the focus of this paper, that active Cas9, coupled with sgRNAs, can bring about transcription inhibition when positioned near the transcription start site, and at the same time bring about cleavage at a second set of sites targeted using regular length gRNAs. This paper also shows that if a Cas9-VPR fusion is used transcriptional activation can be brought about using the sgRNAs in *Drosophila*.

Thus, the novel contribution of this paper is to show that transcriptional inhibition at one site can be coupled with cleavage at another through gRNA multiplexing and manipulation of gRNA length. In mammals this (cleavage at one site and inhibition at another) could probably be achieved using a Cas9-Krab domain fusion, in which the Krab domain acts to inhibit transcription. But, in insects, the Krab domain strategy does not work and there is a need for another approach.

The authors also make a good case for why cleavage at one site and inhibition of gene expression at another would be interesting, in the context of gene drive.

The data throughout the paper is clear and convincing. In the below I focus primarily on a few questions I had on the text and figures, and thinking about how the system can fail, and alternative strategies for achieving the same end. In the context of these thoughts, I make some suggestions as to how the authors might expand the discussion a bit, highlight some potential caveats/challenges, and point to future research directions. These are all just suggestions. Since the paper is not long the discussion provides a real opportunity to flesh things out a bit more.

1. In the introduction where the authors are summarizing some previous literature, they say that inhibition using truncated gRNAs has only been shown on synthetic promoters

"Further modified Cas9 forms

74 fused to activation or repression domains and engineered tgRNAs have been employed

75 to achieve transcriptional activation of endogenous genes or repression of synthetic

76 promoters (15,17)."

A quick medline search using Cas9 and Krab makes shows a lot of papers. Many seem to have used dCas9 to target endogenous promoters. I would just ask the authors to check through these to confirm that none used the tgRNA and active Cas9 to target endogenous promoters. If this is the point the authors want to make (it has only been shown to work on engineered rather than endogenous), maybe they could say this explicitly, just to make the missing piece clear.

2. Missing the word "of" after generation

152 Having demonstrated efficient and penetrant generation LOF phenotypes with both full153

3. In the following I am just curious why only half the progeny are phenotypically ebony. This is not a critical point, but I would have guessed that males carrying vasa-cas9 and two full length gRNAs against ebony, when crossed to ebony homozygous females, would have resulted in a very high frequency of ebony progeny (based on high frequency male germline cleavage and inactivation), unless it just happens that a significant number of the cleavage and repair mechanisms result in a functional protein. I just want to make sure I am not missing something here.

"crossing F1

157 males carrying both vasa-cas9 and full-length gRNAs to reference y or e mutants and

158 then scoring F2 offspring for germline transmission of y or e NHEJ-induced

159 mutations. We found that males carrying both the vasa-cas9 and gRNAs-e(1+2)

160 transgenes resulted in approximately half of their F2 progeny (average = 50%) exhibiting

161 full-bodied e mutant phenotypes (Fig. 2a)"

4. The authors note that cleavage of the multiple yellow loci present in the transheterozygous males results in male sterility, so progeny cannot be scored for inheritance of yellow. The authors put this down (perhaps correctly) to aneuploidy that results from cleavage at not only the endogenous locus (two sites) but also two other copies of yellow present in transgenes. They dont need to comment further on this,but might consider doing so in the discussion. Complete sterility is very different from partial sterility, and given the modest number of sites cleaved, and the fact that none of them are in essential genes,its just something that might suggest something more interesting is going on.

5. The authors go on to show that single tgRNAs can mediate transcriptional suppression, but I think the data here and later also make it clear that we do not really know the rules at TATA promoters or others, exactly where to put these elements to guarantee suppression. In the discussion the authors make a nod to this point in the following

"Given that not all tgRNAs are

351 efficient, a good general strategy for optimizing the likelihood of success would be to

352 express two sgRNAs per target gene from a single plasmid such as the convenient vector"

and here

"365 overexpression. Also, if rules can be inferred for which tgRNAs are most likely to be
366 effective, tgCRISPRi could potentially be exploited for genome-scale transgenic..."

These are appropriate. The authors might also consider just making at one or the other place a more explicit statement that the rules determining which tgRNAs/positions work are unknown and need to be worked out. One thing they might consider noting is that it is possible that some of them just do not bind to DNA well, perhaps based on nucleosome positioning or some other chromatin feature. In short, it may not be the position alone with respect to TSS that is determining success or failure.

6. The gene drive experiment is straightforward and tests the simple question of whether homing levels (mediated by full length gRNAs that bring about cleavage) are influenced by the presence of additional tgRNAs that target Cas9 elsewhere and cannot support cleavage. Importantly, they find that there is no effect on homing rates. This fits with some earlier data from our lab (Oberhofer and colleagues, PNAS 2020) showing that the presence of 4 gRNAs that do not participate in drive (in this case a cleave and rescue drive not utilizing homing) does not interfere with the activity of 4 others that do. Both sets of observations tell us that the levels of gRNA-loaded Cas9 are not limiting for drive. It might be useful for the authors to make this point explicitly – that levels of homing could have been lower if levels of active Cas9/gRNA were limiting. Their results show they are not. Therefore, other factors must be controlling homing/drive frequency, and the "extra" Cas9 available can be used to do other things.

7. Finally, in the context of gene drive the authors note that the above data support strategies in which homing drive (or even ClvR or any other drive mechanism that is cleavage mediated) is used to carry a cargo (tgRNAs) that are used to silence or activate other genes that promote a desirable phenotype such as inability to transmit disease. This is all true.

However, it would be worth noting some related strategies and some caveats/unknowns that remain to be explored.

1. one could achieve the same inhibition of an effector gene if the homing element carried a polII promoter driving siRNA or miRNAs targeting the locus of interest in a tissue specific manner. Of course, this requires that RNAi be efficient in the target tissue, and so on. Just something to note.

2. In order for the authors approach to work (tgRNA-mediated suppression of a gene in the midgut, for example) Cas9 would need to be expressed in the somatic tissue of interest. Since the gRNAs are typically expressed under the control of ubiquitous polIII promoters this means cleavage at the homing locus will be happening here as well. This may not be a problem, but it might, depending on what locus the element is being homed into. Its just something to note. That there is an additional requirement. And given the current obsession with bringing about germline specific expression of Cas9 to avoid off target effects during homing, its something that would be good to outline.

3. Finally, as with all sequenced based approaches to modifying a population the issue of sequence polymorphisms in large populations and how they might inhibit gRNA-dependent Cas9 binding always comes up. A population modification drive does not necessarily put a large fitness cost on a population the way a suppression strategy does. Thus, selection for un-bindable sites near promoters of target genes may not be high but there may well be some more modest fitness cost which will bring about selection for un-bindable sites over time. The big question is "what are the levels of existing polymorphisms near promoters and TSSs". My general sense is that these are (thought to be) less constrained than in a coding region of a highly conserved gene. The concern is following. Drive occurs to fixation at the homing locus. In parallel with that the tgRNAs do their job repressing or activating the effector gene in a specific tissue, which is unlinked. If expression of the effector results in some fitness cost, then the version of that gene that is sensitive to the tgRNAs will be selected against in favor of versions that are not bound by the tgRNA complex. This would lead to a population in which drive was complete but expression at the effector locus was

often not modified in a beneficial way. Multiplexing of the tgRNAs can perhaps help with this, as could multiplexing of mechanisms of inhibition (miRNAs and tgRNAs). The point is, there is a problem, but there are also potential solutions.

My point in all of this is I think that while its relatively straightforward (some of the time) to show success of a strategy in the lab, the real test comes when we imagine it out in the wild. In this context the focus needs to be on failure, its inevitability, and how we can delay it as long as possible. I think readers would appreciate a discussion that walks them through what this would look like.

Bruce Hay

Reviewer #3:

Remarks to the Author:

Summary:

In their manuscript, tgCRISPRi: Efficient gene knock-down using truncated gRNAs and catalytically active Cas9, Auradkar et al. explore the use of truncated gRNAs (tgRNAs) with active Cas9 to inhibit gene expression. They convincingly demonstrate that tgCRISPRi is effective in knocking down expression by ~75% without cleaving DNA. They go on to demonstrate that tgCRISPRi can be used in combination with standard gene drives and that it works similarly well in HEK cells to target a transgene. However, it is unclear if tgCRISPRi offers significant advantages over existing CRISPRi approaches that will lead to widespread adoption.

Key concerns:

1. The primary concern is the lack of comparison to existing CRISPRi approaches, e.g., full-length gRNAs with catalytically dead Cas9 (Ghosh et al., 2016, which should also be cited). Throughout, comparisons are instead made to traditional CRISPR (full-length gRNAs + catalytically active Cas9), which isn't the relevant comparison. Rather than comparing to a knock-out approach, tgCRISPRi should be compared to other knock-down approaches so potential adopters can weigh the pros and cons of different methods.

The only comparison to CRISPRi I noted is presented in Figure S1 without any numbers. Although it's hard to say absent quantitative comparisons, based on the statement in the text that they observe "similar LOF phenotypes," it doesn't appear as though tgCRISPRi is more effective than CRISPRi.

2. The conclusions are often qualitative and Ns are not always provided (e.g., Figs 1 and S1). Individual data points should be shown for all experiments (e.g., Figs. 2E and S5).

3. The conclusion that targeting sites between the TATA box and TSS initiator region is most effective is based on a very small number of targets. This statement (lines 212-214) should be qualified accordingly.

Note to all reviewers regarding addition of new figure (Figure 6) to revised manuscript

While it is our firm opinion that data provided in the original manuscript is novel and of broad interest to the *Drosophila* and broader CRISPR communities (with Reviewers 2 and 3 concurring), we have now added yet another application of the tgRNA approach to the revised manuscript that further increases its novelty. In these experiments, presented in the newly added Figure 6, we show that by targeting conserved transcription factor binding sites in cis-regulatory modules (CRMs, or enhancers), we can either reduce gene activity (shown for the Sex combs reduced T1 CRM and the *knirps* L2-vein CRM) or activate gene expression (shown for the *knirps* L2-vein CRM). This latter example also highlights an advantage particular to tgRNA-based silencing to rule out off target effects. In this instance, we demonstrate that the activating effect of the tgRNA not only mimics known effects of ectopic *knirps* expression in the wing, but does so via its action of the L2-CRM. For this incisive experiment, we make use of an orthologous L2-CRM from a related *Drosophila* species (*D. grimshawi*) that can substitute functionally for the *D. mel* L2-CRM, but lacks the tgRNA target sites present in its *D. mel* counterpart. The *D. grim.* L2-CRM thus provides a negative control to demonstrate that the *knirps* tgRNAs act directly via the L2-CRM and not non-specifically or indirectly by altering some other aspect of the L2 vein gene regulatory network. This example, again highlights the novelty and flexible advantages of the tgCRISPR method. We note that applications of all these new tools extend beyond gene-drives with broad relevance to diverse CRISPR editing systems.

REVIEWER COMMENTS

Reviewer #1 (Remarks to the Author):

In this study Auradkar et al. use truncated sgRNAs (tgRNAs) in combination with wildtype Cas9 for transcriptional interference of target genes. Working mostly in Drosophila, the authors show that such a CRISPRi system can repress the expression of four endogenous target genes to sufficient levels to observe loss-of-function phenotypes. They also demonstrate that tgRNAs can be combined with a conventional nuclease driven gene drive and demonstrate inhibition of a transgenic reporter in HEK293T cells.

While the experiments described in this manuscript are sound and the data support most of the conclusions, I do not believe that this study presents a sufficient conceptual advance or a method of broad enough interest to warrant publication in Nature Communications. I will substantiate this view in the major points below.

Major points:

1. *The tgCRISPRi system does not present a significant conceptual advance. The fact that truncated sgRNAs can mediate binding of Cas9 to the target locus without activating nuclease activity has been demonstrated on multiple occasions and that recruiting Cas9 to the target locus without cutting it can elicit a CRISPRi effect is equally well known. Moreover, previous studies have directly demonstrated that combining tgRNAs and Cas9 can be used to modulate transcription.*

- A. *Santos-Moreno et al. and Jeong et al. both have shown that truncated sgRNAs can mediate CRISPRi in prokaryotes (PMID: 36787424, PMID: 32488086).*

We thank the reviewer for bringing our attention to this paper. We have added the recommended citation in referencing CRISPRi studies using shortened gRNAs in prokaryotes. See line 76, citation no. 21, 22. We note that the authors of this study employed dCas9, not active Cas9, to inhibit gene expression.

- B. *Furthermore, the very first description of CRISPRi in mammalian cells already tested truncated sgRNAs and showed that guides shortened by up to 8 nucleotides can mediate gene repression (PMID: 33545038).*

We have also added this citation along with the one above, but note that this study only examined the activity of shortened gRNAs with dCas9 only in prokaryotes. See line 76, citation no. 18.

- C. *Maybe most significantly, Ye et al. have used tgRNAs and catalytically active Cas9, the exact method described here, for CRISPRi in mammalian cells (PMID: 30062046).*

*We again thank the reviewer for pointing out this study, in which a shortened gRNA was used in combination with an MS2 loop to effectively repress the target genes using the Krab transcriptional repression domain with active Cas9. This strategy led to a maximum of two-fold reduction in gene repression. Our tgRNA strategy is simpler and yields much more efficient suppression, without the use of a specialized fusion. For these reasons, the tgRNA strategy has potential for combinatorial approaches. In addition, the Krab domain strategy has been shown not be effective in *Drosophila* (see comment from Reviewer 2 raising this issue in support of our approach that relies on an entirely wild-*

type form of Cas9). Therefore, as pointed out by Reviewer 2, we need alternative strategies to mediate efficient gene knockdown in *Drosophila*. The tgCRISPRi system we demonstrate to be effective in this study can thus provide a novel solution by using tgRNAs targeting sites near the TSS to mediate efficient target gene knockdown combined with active Cas9. We have added reference to Ye et al. as citation no. 20, see line 75 .

D. *In summary, iterations of the tgCRISPRi system have already been described on multiple occasions. The fact that none of the references listed above are currently cited in the manuscript suggests that the authors could improve the referencing of prior work.*

As indicated above we appreciate the new references kindly provided by the reviewer. We note, however, that all the prior studies we have cited in the manuscript as well as these new citations pertain to the use of shortened gRNAs (e.g., similar to tgCRISPRi system) with a catalytically active form of Cas9 in prokaryotes, or employed a form of KRAB fusion in mammalian cells. As we elaborate further in response, to the next point below, our primary goal in this study was to develop an add-on tool to improve the effectiveness of gene-drives, which also has broader applications to other gene-editing scenarios in which it is advantageous to employ an active form of Cas9 while also modifying expression of secondary loci.

2. The tgCRISPRi system is unlikely to be a method of widespread interest. This paper mainly describes the use of tgCRISPRi in *Drosophila*.

We respectfully disagree with the reviewer on this point.

Importantly, we wish to clarify a key point that we may not have stated adequately in the original manuscript, namely, that a primary goal in developing this first tgRNA system in *Drosophila* was to create an add-on to gene-drive systems, which rely on fully active Cas9. Such combination of Cas9 driving (with a full-length gRNA) and a tgRNA to either repress or activate secondary target loci would have significant value. We explain in this context that such a goal cannot easily be accomplished either by using full-length gRNAs to mutagenize auxiliary targets (due to mutation load and toxicity) or by standard CRISPRi that relies on full-length gRNAs and a dCas9-KRAB fusion protein, which by its nature could not sustain the drive activity. We have revised the text to emphasize this important point and our motivating reasons for conducting this study.

Thus, the proof-of-concept demonstration of tgCRISPRi we present in this manuscript should have a substantial impact in the gene drive field (in any organism), with particular emphasis on its wide-ranging applications in controlling plant pests and diseases vectored by insects such as mosquitoes. We cite four specific examples in the revised discussion to highlight this important point:

- 1) It should be possible to employ tgRNAs to improve the efficiency of gene-drives by suppressing genes that associated with NHEJ repair pathway or by activating expression of components which promote HDR. Alternatively, by using modified tgRNAs which include a MS2 scaffold to recruit VPR, genes encoding HDR pathway components could be overexpressed to potentially increase gene drive efficiency.
- 2) tgRNAs also have the potential to target crucial mosquito/host genes that are necessary for the survival of malarial parasites. This could involve suppressing expression of host genes such as FREP1, Ctl5, TEPs, that are required for parasite maturation, or by enhancing expression of immunity factors such as Rel1, APL1C, LRIM1 to obstruct parasite growth in mosquitoes.
- 3) The tgCRISPR system could also be used to suppress pest populations by incorporating tgRNA that targets sex-specific genes into a gene drive. This will effectively repress the female population and lead to overall population suppression.
- 4) We have also enhanced the novelty of the study further by adding new data to the manuscript that demonstrate the novel possibility of targeting cis-regulatory modules (CRMs) or enhancer regions by tgRNAs either to reduce or increase gene activity in specific tissues or cell types (new Figure 6). This original application of the tgCRISPRi technology, which to our knowledge has not been shown in any system, provides a much greater degree of precision in modulating target gene expression since it selectively targets only one component of gene activity. Targeting CRMs should also prove particularly useful for analysis of developmental processes which are often regulated in a tissue-specific pattern via dedicated enhancer elements. Moreover, the use of tgRNAs targeting enhancer regions could facilitate population replacement without causing any DNA damage, by replacing an existing insect population with one carrying a modified tgRNA binding site that no longer interacts with the Cas9 and thus does not suffer fitness costs associated with gene mis-regulation mediated by the Cas9 (new Figure 6i).

In addition, due to its simplicity compared to published CRISPRi methods, we anticipate that tgRNA strategies may represent a useful addition to the expanding CRISPR toolkit for gene editing strategies as well.

3. CRISPRi is relatively under-explored in flies, almost certainly because the *Drosophila* community already has an excellent system for gene knockdown in the form of RNAi. There are several large-scale transgenic resources for RNAi in flies that are publicly available and together cover almost the entire genome. They are undoubtedly one of the most widely used tools in the *Drosophila* community. Although RNAi works very well in flies compared to other systems, it is not without its drawbacks. Most importantly, gene knockdown is typically incomplete and there is a significant fraction of RNAi lines where knockdown is so inefficient that no phenotype can be observed (PMID: 26320097). However, the tgCRISPRi system suffers from the exact same drawbacks. The knockdown efficiencies of the tgRNAs tested in this study are well within the range typically observed with RNAi (PMID: 26320097) and also include a significant proportion of inactive tgRNAs. Thus, this new method essentially does what current state-of-the-art RNAi does without improving on its limitations. Off-target effects are another limitation of RNAi, but the current manuscript does not test for on-target specificity of tgCRISPRi.

For this reason, it is hard to see why many people would go through the trouble of generating new tgRNA transgenic lines when there are already RNAi lines that can be used for the same purpose. There is more utility for this system in mammalian cells, where RNAi is more problematic, but as outlined above, tgCRISPRi has already been described in that system.

We acknowledge the reviewer's point that RNAi is a well-established method for gene knockdown in the *Drosophila* community. We note, however, that tgCRISPRi is considerably less complicated to include in a construct than RNAi, which typically requires high levels of expression, subsequently often requiring multiple components for providing such strong expression (e.g., an amplifying GAL4 source and an UAS-RNAi cassette). In the case of tgCRISPR, only the tgRNA needs to be added to the gene-drive cassette. Also, the tgCRISPRi system could readily be adapted to support gene-drives in other insect species where RNAi may be less developed or tested, mosquitoes being one clear example. Furthermore, due to the requirement for precise positioning of the tgRNA relative to the TSS of target genes to repress gene expression, off-target binding of tgRNA is expected to have negligible effects on non-target genes, which is a considerable concern for RNAi. Therefore, for these various and synergistic reasons, the tgCRISPRi system should prove to be a valuable novel tool for gene drive community and should expand realm of organisms for which gene knock-down technology is available. We discuss this point in the revised manuscript. Line 470-479.

4. At the beginning of the manuscript the authors compare the phenotypes caused by tgCRISPRi targeting *yellow* and *ebony* with those caused by Cas9-mediated indel mutagenesis using full-length sgRNAs targeting the same target sites. This comparison can be misleading, as sgRNAs for CRISPRi/a or CRISPRn should be designed differently. While guides for transcriptional interference should be targeted near the transcriptional start site (as done here), sgRNAs for gene disruption with CRISPR nucleases should be targeted to the coding sequence, where out-of-frame indels are likely to disrupt gene function. Therefore, the experiments in this manuscript compare correctly designed tgCRISPRi guides to incorrectly designed CRISPRn guides. I see the point of doing these experiments to demonstrate the effect at the on-target sites at sequence level, but with regard to phenotypic penetrance, readers would need to be alerted in the main text that this constitutes comparing apples with oranges. Furthermore, the statement in lines 87-89 should be removed for the same reason.

As recommended, we conducted additional experiments to compare the effectiveness of standard CRISPR mutagenesis (active Cas9 + full length gRNA), CRISPRi (dCas9 + full length gRNA), tgCRISPRi (active Cas9 + shortened tgRNA) and tgCRISPRi (dead Cas9 + shortened tgRNA). The findings have been included in Figure 1c and Sup Fig. 1a. The overall conclusion is the same as previously, namely that tgCRISPRi is comparable in efficiency to the other strategies, thus avoiding the toxicity associated with mutating multiple targets with standard CRISPR while permitting efficient scarless gene knock-down in the presence of fully active Cas9.

These clarifying points have been made in the revised text. Line 155-161, Line 210-216.

5. The amount of knock-down necessary to generate a loss-of-function phenotype is highly gene specific (e.g. *ebony* is known to be partially haploinsufficient).

We agree with the reviewer that *ebony* is known to be partially haploinsufficient. In addition to phenotype data, we have conducted an RT-PCR analysis to quantify the efficiency of tgRNA knockdown, which ranges from 70-90%. See Figure 2e, Supplementary Figure 1a, and 6a for further details. We also note that the phenotype of tgRNAe expressing flies is indistinguishable from null homozygous mutants, which have a much darker and uniform pigmentation phenotype than heterozygotes.

6. In contrast, quantification of transcript abundance is a gene agnostic measure on the efficiency of tgCRISPRi. It would strengthen the manuscript if it were performed on all targets.

During pupal stages, the expression patterns of the *yellow* and *wingless* gene are highly tissue specific and dynamic, making reliable quantification by RT-PCR problematic. In order to accurately quantify transcript levels, we confined our RT-PCR quantification specifically to the more ubiquitously expressed *ebony* and *white* genes. As the reviewer correctly pointed out in a previous comment, the gene for which this quantification is the most informative is *ebony* due to its moderate haploinsufficient phenotype. See Figure 2e, Supplementary Figure 1a, and 6a-c for RT-PCR quantification of *ebony* and *white* genes.

Minor points:

7. Consider changing CRISPR/Cas to CRISPR-Cas.

We have made the recommended change.

8. Line 132: The target sites are not in the reading frame.

We have made the recommended change.

9. Line 203: - 129 not + 129

We have made the recommended change.

10. Line 340: As outlined above the tgCRISPRi system has been described before.

We have further clarified that this is the first application of tgCRISPRi to *Drosophila*, although to be precise, our use of a fully active unmodified form of Cas9 together with a tgRNA in mammalian cells is unique to our knowledge since prior experiments using shorted gRNAs employed either dCas9 or a form of Cas9 fused to either a KRAB domain (CRISPRi) or an activation domain fused to Cas9 (e.g. Cas9-VP16).

11. Line 350 and line 374: remove “fully” and “complete (100%)”. All shown tgRNAs lead to knock-downs of target gene expression of no more than 80%.

We have made the recommended change.

12. Figure 5b: The pigmentation phenotype is hard to see, but there are clear signs of toxicity (fusion defects in thorax and abdomen). Please discuss this in the results section.

We observed that overexpression of *ebony* using *pnr G4>Cas9VPR* resulted in subtle but reproducible pigmentation phenotypes similar to those shown in a prior published study using *pnrG4>UAS-ebony* (PMID: 21878507). That study also reported some toxicity in the thorax due to overexpression of UAS-*ebony*, which we also observe. We discuss this point in the revised manuscript.

Reviewer #2 (Remarks to the Author):

This is an interesting manuscript in which the authors explore what can be done with truncated gRNAs alone, or in combination with long gRNAs, all in the context of Cas9.

*As background, it has been shown earlier that long gRNAs (called just gRNAs from here out) bound to dead Cas9 can inhibit transcription when bound near a transcription start site. It has also been shown that truncated gRNAs (tgRNAs) bound to active Cas9 cannot activate cleavage but can, in the context of a Cas9-VPR fusion protein, activate transcription (though this has not, until this paper, been shown in *Drosophila*). In this latter context (mammalian cells) it has also been shown that this fusion can do two things at once, cleave one set of sites and activate transcription from others.*

*What has not been shown (I think) is the focus of this paper, that active Cas9, coupled with sgRNAs, can bring about transcription inhibition when positioned near the transcription start site, and at the same time bring about cleavage at a second set of sites targeted using regular length gRNAs. This paper also shows that if a Cas9-VPR fusion is used transcriptional activation can be brought about using the sgRNAs in *Drosophila*.*

Thus, the novel contribution of this paper is to show that transcriptional inhibition at one site can be coupled with cleavage at another through gRNA multiplexing and manipulation of gRNA length. In mammals this (cleavage at one site and inhibition at another) could probably be achieved using a Cas9-Krab domain fusion, in which the Krab domain acts to inhibit transcription. But, in insects, the Krab domain strategy does not work and there is a need for another approach.

The authors also make a good case for why cleavage at one site and inhibition of gene expression at another would be interesting, in the context of gene drive.

The data throughout the paper is clear and convincing. In the below I focus primarily on a few questions I had on the text and figures, and thinking about how the system can fail, and alternative strategies for achieving the same end. In the context of these thoughts, I make some suggestions as to how the authors might expand the discussion a bit, highlight some potential caveats/challenges, and point to future research directions. These are all just suggestions. Since the paper is not long the discussion provides a real opportunity to flesh things out a bit more.

We thank the reviewer for these supportive and thoughtful comments, which we address more specifically below.

1. In the introduction where the authors are summarizing some previous literature, they say that inhibition using truncated gRNAs has only been shown on synthetic promoters

"Further modified Cas9 forms fused to activation or repression domains and engineered tgRNAs have been employed to achieve transcriptional activation of endogenous genes or repression of synthetic promoters (15,17)."

A quick medline search using Cas9 and Krab makes shows a lot of papers. Many seem to have used dCas9 to target endogenous promoters. I would just ask the authors to check through these to confirm that none used the tgRNA and active Cas9 to target endogenous promoters. If this is the point the authors want to make (it has only been shown to work on engineered rather than endogenous), maybe they could say this explicitly, just to make the missing piece clear.

We have made the recommended clarifying changes to enhance the text. As indicated in our response to Reviewer 1, we have added citations of additional relevant work, which include referencing research on prokaryotes that utilized tgRNA and active Cas9 to target bacterial genes. Additionally, we have added a citation to a study using shortened gRNAs in mammalian cells, which involved the use of a Krab domain to suppress expression on endogenous promoters. Line 70-78.

2. Missing the word "of" after generation

152 Having demonstrated efficient and penetrant generation LOF phenotypes with both full153

We have made the recommended change.

3. In the following I am just curious why only half the progeny are phenotypically ebony. This is not a critical point, but I would have guessed that males carrying vasa-cas9 and two full length gRNAs against ebony, when crossed to ebony homozygous females, would have resulted in a very high frequency of ebony progeny (based on high frequency male germline cleavage and inactivation), unless it just happens that a significant number of the cleavage and repair mechanisms result in a functional protein. I just want to make sure I am not missing something here.

"crossing F1 males carrying both vasa-cas9 and full-length gRNAs to reference y or e mutants and then scoring F2 offspring for germline transmission of y or e NHEJ-induced mutations. We found that males carrying both the vasa-cas9 and gRNAs-e(1+2) transgenes resulted in approximately half of their F2 progeny (average = 50%) exhibiting full-bodied e mutant phenotypes (Fig. 2a)"

This is an interesting question for which there may be several possible explanations. First, because the gRNAs target non-coding sequences close to TSS, it is possible that a fraction of the cleaved and imprecisely repaired alleles may permit some level of functional protein expression, although this should not be too common given the full body ebony phenotypes of the F1 males. Second, it is also possible that there are some transvection-related effects although less severe (e.g., non-deletion class) alleles generated by the gRNAs that permit regulatory sequences from the reference allele to drive expression of the functional ebony coding sequences from the gRNA-mutated promoter alleles. The main point of this experiment, however, was simply to show that NHEJ mutations were generated as expected when using full-length gRNAs but not when using truncated tgRNAs

4. The authors note that cleavage of the multiple yellow loci present in the transheterozygous males results in male sterility, so progeny cannot be scored for inheritance of yellow. The authors put this down (perhaps correctly) to aneuploidy that results from cleavage at not only the endogenous locus (two sites) but also two other copies of yellow present in transgenes. They dont need to comment further on this,but might consider doing so in the discussion.

Complete sterility is very different from partial sterility, and given the modest number of sites cleaved, and the fact that none of them are in essential genes,its just something that might suggest something more interesting is going on.

This is an interesting point and we have added a sentence to the manuscript raising the possibility that this sterile phenotype might warrant further examination. Line 184-185.

5. The authors go on to show that single tgRNAs can mediate transcriptional suppression, but I think the data here and later also make it clear that we do not really know the rules at TATA promoters or others, exactly where to put these elements to guarantee suppression. In the discussion the authors make a nod to this point in the following "Given that not all tgRNAs are efficient, a good general strategy for optimizing the likelihood of success would be to express two sgRNAs per target gene from a single plasmid such as the convenient vector" and here overexpression. Also, if rules can be inferred for which tgRNAs are most likely to be effective, tgCRISPRi could potentially be exploited for genome-scale transgenic..."

These are appropriate. The authors might also consider just making at one or the other place a more explicit statement that the rules determining which tgRNAs/positions work are unknown and need to be worked out. One thing they might consider noting is that it is possible that some of them just do not bind to DNA well, perhaps based on nucleosome positioning or some other chromatin feature. In short, it may not be the position alone with respect to TSS that is determining success or failure.

We appreciate this point and have added a statement mentioning this possibility to the discussion. Line 447-453.

6. The gene drive experiment is straightforward and tests the simple question of whether homing levels (mediated by full length gRNAs that bring about cleavage) are influenced by the presence of additional tgRNAs that target Cas9 elsewhere and cannot support cleavage. Importantly, they find that there is no effect on homing rates. This fits with some earlier data from our lab (Oberhofer and colleagues, PNAS 2020) showing that the presence of 4 gRNAs that do not participate in drive (in this case a cleave and rescue drive not utilizing homing) does not interfere with the activity of 4 others that do. Both sets of observations tell us that the levels of gRNA-loaded Cas9 are not limiting for drive. It might be useful for the authors to make this point explicitly – that levels of homing could have been lower if levels of active Cas9/gRNA were limiting. Their results show they are not. Therefore, other factors must be controlling homing/drive frequency, and the "extra" Cas9 available can be used to do other things.

We have added the recommended citation. Line 489-494, citation 50.

7. Finally, in the context of gene drive the authors note that the above data support strategies in which homing drive (or even ClvR or any other drive mechanism that is cleavage mediated) is used to carry a cargo (tgRNAs) that are used to silence or activate other genes that promote a desirable phenotype such as inability to transmit disease. This is all true.

However, it would be worth noting some related strategies and some caveats/unknowns that remain to be explored.

We have added sections to the discussion raising some of these interesting points as indicated below. Line 505-526.

1. one could achieve the same inhibition of an effector gene if the homing element carried a polIII promoter driving siRNA or miRNAs targeting the locus of interest in a tissue specific manner. Of course, this requires that RNAi be efficient in the target tissue, and so on. Just something to note.

While use of genome-wide GAL4 driven UAS-RNAi lines have proven invaluable for research in *Drosophila*, we note that such comprehensive resources do not exist for other systems such as mosquitoes. In such scenarios tgCRISPRi may prove to be a particularly valuable alternative for gene suppression. Also, RNAi typically requires fairly high levels of expression that are readily generated by the GAL4-UAS system, which would be bulky to include as an auxiliary element, although in some contexts RNAi constructs could be effective when expressed directly by a pol-II promoter. A question to address in the future is whether tgCRISPR will prove to have fewer off-target effects than occur with RNAi, which in some contexts can pose significant challenges.

2. In order for the authors approach to work (tgRNA-mediated suppression of a gene in the midgut, for example) Cas9 would need to be expressed in the somatic tissue of interest. Since the gRNAs are typically expressed under the control of ubiquitous polIII promoters this means cleavage at the homing locus will be happening here as well. This may not be a problem, but it might, depending on what locus the element is being homed into. Its just something to note. That there is an additional requirement. And given the current obsession with bringing about germline specific expression of Cas9 to avoid off target effects during homing, its something that would be good to outline.

Finally, as with all sequenced based approaches to modifying a population the issue of sequence polymorphisms in large populations and how they might inhibit gRNA-dependent Cas9 binding always comes up. A population modification drive does not necessarily put a large fitness cost on a population the way a suppression strategy does. Thus, selection for un-bindable sites near promoters of target genes may not be high but there may well be some more modest fitness cost which will bring about selection for un-bindable sites over time. The big question is "what are the levels of existing polymorphisms near promoters and TSSs". My general sense is that these are (thought to be) less constrained than in a coding region of a highly conserved gene. The concern is following. Drive occurs to fixation at the homing locus. In parallel with that the tgRNAs do their job repressing or activating the effector gene in a specific tissue, which is unlinked. If expression of the effector results in some fitness cost, then the version of that gene that is sensitive to the tgRNAs will be selected against in favor of versions that are not bound by the tgRNA complex. This would lead to a population in which drive was complete but expression at the effector locus was often not modified in a beneficial way. Multiplexing of the tgRNAs can perhaps help with this, as could multiplexing of mechanisms of inhibition (miRNAs and tgRNAs). The point is, there is a problem, but there are also potential solutions.

We address this very valid point the following way: "Another consideration is that if the tgRNA produces a fitness cost then when combined with a drive it might eventually lead to the evolution of target site mutations that render it inactive. A potential solution to this problem would be to employ two or more tgRNAs that efficiently act on a given target gene to provide redundancy in target gene modulation."

My point in all of this is I think that while its relatively straightforward (some of the time) to show success of a strategy in the lab, the real test comes when we imagine it out in the wild. In this context the focus needs to be on failure, its inevitability, and how we can delay it as long as possible. I think readers would appreciate a discussion that walks them through what this would look like.

We appreciate these insightful reflections on the potential caveats inevitably associated with gene-drive implementation in the wild.

Bruce Hay

Reviewer #3 (Remarks to the Author):

Summary:

In their manuscript, tgCRISPRi: Efficient gene knock-down using truncated gRNAs and catalytically active Cas9, Auradkar et al. explore the use of truncated gRNAs (tgRNAs) with active Cas9 to inhibit gene expression. They convincingly demonstrate that tgCRISPRi is effective in knocking down expression by ~75% without cleaving DNA. They go on to demonstrate that tgCRISPRi can be used in combination with standard gene drives and that it works similarly well in HEK cells to target a transgene. However, it is unclear if tgCRISPRi offers significant advantages over existing CRISPRi approaches that will lead to widespread adoption.

Key concerns:

1. The primary concern is the lack of comparison to existing CRISPRi approaches, e.g., full-length gRNAs with catalytically dead Cas9 (Ghosh et al., 2016, which should also be cited). Throughout, comparisons are instead made to traditional CRISPR (full-length gRNAs + catalytically active Cas9), which isn't the relevant comparison. Rather than comparing to a knock-out approach, tgCRISPRi should be compared to other knock-down approaches so potential adopters can weigh the pros and cons of different methods.

We agree with the reviewer and now endeavor to clarify our primary point in this regard, namely that in for a gene-drive, or for other contexts where one may be tied to using an active form of Cas9, the tgRNA system offers an effective approach to modulating expression of secondary gene targets. In such cases, using a full-length gRNA to mutate such secondary target genes (if one wished to reduce their function), great fitness costs could often be associated with the genome damage such cleavage events would generate. The only alternative in such cases, would be RNAi knock-down, which in Drosophila is very well supported by existing genome-wide resources, but in other species without such resources tgRNAs might prove to be a useful alternative. Also, in many cases, RNAi works best when high expression levels of the interfering RNA are generated using the GAL4-UAS system, which may not be so easy to achieve via direct enhancer-RNAi fusion genes. Incorporating a tgRNA into a gene drive would be simpler and space saving compared to adding both GAL4 and UAS-RNAi constructs to a drive cassette. We have added discussion of some of these points as also suggested by reviewers 1 and 2.

We have also added the recommended citation. Citation 10.

The only comparison to CRISPRi I noted is presented in Figure S1 without any numbers. Although it's hard to say absent quantitative comparisons, based on the statement in the text that they observe "similar LOF phenotypes," it doesn't appear as though tgCRISPRi is more effective than CRISPRi.

We have performed recommended experiment and results are presented in Figure 1c and S1a.

2. *The conclusions are often qualitative and Ns are not always provided (e.g., Figs 1 and S1). Individual data points should be shown for all experiments (e.g., Figs. 2E and S5).*

We have added Ns for required figures. We have also added data points to all the graphs.

3. *The conclusion that targeting sites between the TATA box and TSS initiator region is most effective is based on a very small number of targets. This statement (lines 212-214) should be qualified accordingly*

We have clarified the statement.

Reviewers' Comments:

Reviewer #1:

Remarks to the Author:

I would like to thank the authors for their thoughtful response to my comments and the resulting changes to the manuscript. The authors have made several changes to the text, which now more comprehensively cites prior work and better explains the rationale for this study. In addition, they have added some new data. There is now a comparison of tgCRISPRi to conventional CRISPRi, which shows that these techniques function with similar efficiency. In addition, there is a new figure showing gene regulation at the level of individual enhancers. This is a nice proof of concept for using CRISPR to fine tune gene expression, although it almost certainly is not specific to the described tgCRISPR technique, as similar results would be expected with conventional CRISPRi/a.

My main concern with this study has been that it is unlikely to be of broad interest to scientists working with the *Drosophila melanogaster* system, as large-scale resources for similar techniques (RNAi, CRISPRi/a) exist. This sentiment was echoed by reviewer 3, who also questioned whether tgCRISPR offers significant advantages over established CRISPRi approaches. The new data in the revised manuscript confirms that these methods function with similar efficiency. In their rebuttal letter and in the revised manuscript the authors better explain their motivation for this study. They seem to concur that applications in the model system *D. melanogaster* are likely limited by the wealth of preexisting resources, but point to applications in organisms where such tools do not exist or to the use in gene drives, which necessitate the use of catalytically active Cas9. I agree that these are likely scenarios where the tgCRISPR technique could prove very useful. It is unfortunate that these applications are not better demonstrated in the manuscript. There is a proof of principle experiment showing that a truncated sgRNA does not interfere with a Cas9 gene drive, but this experiment is performed in *D. melanogaster* and uses the yellow and ebony genes, which are not of practical relevance for this application. An experiment that uses tgCRISPR to target components of NHEJ or HDR, thereby possibly increasing drive efficiency, or in an organism of practical relevance for gene drives (e.g. mosquitos), would have gone a long way to directly demonstrate the usefulness of this approach. In its absence this is a solid and interesting proof of principle study that fails to directly demonstrate the applications it promises to deliver.

Reviewer #2:

Remarks to the Author:

The authors have addressed all my concerns.

Bruce Hay

Reviewer #3:

Remarks to the Author:

The authors have addressed my concerns.

In the updated Figure S1, the figure and legend state that dCas9 was used to drive both gRNAs and tgRNAs, whereas the text (lines 210-216) notes that Cas9 was used. I assume the figure is incorrect and should be corrected.

REVIEWERS' COMMENTS

Reviewer #1 (Remarks to the Author):

I would like to thank the authors for their thoughtful response to my comments and the resulting changes to the manuscript. The authors have made several changes to the text, which now more comprehensively cites prior work and better explains the rationale for this study. In addition, they have added some new data. There is now a comparison of tgCRISPRi to conventional CRISPRi, which shows that these techniques function with similar efficiency. In addition, there is a new figure showing gene regulation at the level of individual enhancers. This is a nice proof of concept for using CRISPR to fine tune gene expression, although it almost certainly is not specific to the described tgCRISPR technique, as similar results would be expected with conventional CRISPRi/a.

*My main concern with this study has been that it is unlikely to be of broad interest to scientists working with the *Drosophila melanogaster* system, as large-scale resources for similar techniques (RNAi, CRISPRi/a) exist. This sentiment was echoed by reviewer 3, who also questioned whether tgCRISPR offers significant advantages over established CRISPRi approaches. The new data in the revised manuscript confirms that these methods function with similar efficiency. In their rebuttal letter and in the revised manuscript the authors better explain their motivation for this study. They seem to concur that applications in the model system *D. melanogaster* are likely limited by the wealth of preexisting resources, but point to applications in organisms where such tools do not exist or to the use in gene drives, which necessitate the use of catalytically active Cas9. I agree that these are likely scenarios where the tgCRISPR technique could prove very useful. It is unfortunate that these applications are not better demonstrated in the manuscript. There is a proof of principle experiment showing that a truncated sgRNA does not interfere with a Cas9 gene drive, but this experiment is performed in *D. melanogaster* and uses the yellow and ebony genes, which are not of practical relevance for this application. An experiment that uses tgCRISPR to target components of NHEJ or HDR, thereby possibly increasing drive efficiency, or in an organism of practical relevance for gene drives (e.g. mosquitos), would have gone a long way to directly demonstrate the usefulness of this approach. In its absence this is a solid and interesting proof of principle study that fails to directly demonstrate the applications it promises to deliver.*

We appreciate that Reviewer 1 feels that the revised manuscript is improved and addresses concerns regarding citation of the literature. We also agree with the reviewer that the tgCRISPRi system offers promise for applications such as silencing components of the NHEJ pathway to increase the frequency of gene-conversion for gene-drives. These studies, which would require multigenerational cage trials lie beyond the scope of the current proofs-of-concept study. However, this potential application is suggested for future research in the discussion section.

Reviewer #2 (Remarks to the Author):

The authors have addressed all my concerns.

Bruce Hay

No changes needed.

Reviewer #3 (Remarks to the Author):

The authors have addressed my concerns.

No changes needed.

In the updated Figure S1, the figure and legend state that dCas9 was used to drive both gRNAs and tgRNAs, whereas the text (lines 210-216) notes that Cas9 was used. I assume the figure is incorrect and should be corrected.

We have corrected discrepancy in the text section. To ensure comparable results, we utilized dCas9 to drive both gRNAs and tgRNAs.